# Impact of resolution on Large Eddy Simulation of mid-latitude summer time convection

Christopher Moseley[1,2], Ieda Pscheidt[3], Guido Cioni[1], and Rieke Heinze[1]

[1]Max Planck Institute for Meteorology, Hamburg, Germany
[2]Department of Atmospheric Sciences, National Taiwan University, Taiwan
[3]University of Bonn, Germany

**Correspondence:** Christopher Moseley (christopher.moseley@mpimet.mpg.de)

**Abstract.** We analyze life cycles of summer time moist convection of a Large Eddy Simulation (LES) in a limited area setup over Germany. The goal is to assess the ability of the model to represent convective organization in space and time in comparison to radar data, and its sensitivity to daily mean surface air temperature. A continuous period of 36 days in May and June 2016 is simulated with a grid spacing of 625 m. This period was dominated by convection over large parts of the domain on most of the days. Using convective organization indices, and a tracking algorithm for convective precipitation events, we find that an LES with 625 m grid spacing tends to underestimate the degree of convective organization, and shows a weaker sensitivity of heavy convective rainfall to temperature as suggested by the radar data. An analysis of three days within this period that are simulated with finer grid spacing of 312 m and 156 m showed that a grid spacing at the 100 m scale has the potential to improve the simulated diurnal cycles of convection, the mean time evolution of single convective events, and the degree of convective organization.

## 1   Introduction

An adequate representation of the diurnal cycles of convection in atmospheric models is important for numerical weather prediction and climate simulations, not only for the tropics (Ruppert and Hohenegger, 2018), but also for mid-latitude summertime convection (Pritchard and Somerville, 2009). For this purpose, cloud resolving models (CRMs) without deep cumulus parametrization are increasingly applied thanks to growing computational power. In the meanwhile, first global simulations with grid spacings between 7 km and 2.5 km have been performed (Stevens et al., 2019). This range is usually termed convection *permitting*, as not all relevant processes within convective cells are sufficiently resolved. In fact, in some of these models shallow convection is parametrized in order to correct deficiencies in the simulation of smaller updrafts. Regional limited area models allow for even higher resolutions with grid spacings in the sub-kilometer range with Large Eddy Simulations (LES) where the large eddies of the turbulence spectrum are modeled explicitly as opposed to a fully parametrized turbulence

spectrum in the convection-permitting simulations. Recently, selected diurnal cycles over Germany have been simulated in a realistic LES setup with the model ICON-LEM (Heinze et al., 2017) within the German funded project HD(CP)[2] ("High Definition Clouds and Precipitation for advancing Climate Prediction"). Previous studies have discussed the question which resolution is optimal for a good representation of the processes involved in deep convective updrafts. A semi-idealized study of days with precipitating convection by Petch et al. (2002) with grid spacings between 2 km and 125 m showed that the horizontal resolution should be at least one quarter of the sub-cloud layer depth, and that the best match with observational data was found only at the highest resolution. Similarly, a study by Bryan et al. (2003) showed that for an adequate simulation of a squall line using models with traditional LES closures, grid spacings of the order of 100 m are required. Besides horizontal resolution, there are also other factors that impact the ability of CRMs to simulate convection, such as the subgrid turbulence scheme (Panosetti et al., 2019), the microphysics scheme (Singh and O'Gorman, 2014), and the representation of the land surface.

The formation of strong convective precipitation events depends on several environmental conditions, like air temperature, surface fluxes, large scale forcing, and the ability of convection to organize. The sensitivity of precipitation extremes to warmer temperatures has been heavily discussed in the recent years. The argument that the strongest events should increase at a rate of ca. 7% $K^{-1}$ according to the thermodynamic Clausius-Clapeyron (CC) relation was put forward by Allen and Ingram (2002), and Trenberth et al. (2003). Observational evidence showed that even in mid-latitude regions, these rates can be up to twice the CC rate (Lenderink and Van Meijgaard, 2008; Westra et al., 2014), which is predominantly the case for convective precipitation, while the stratiform precipitation type follows CC more closely (Berg et al., 2013). In the meanwhile, several studies have found a super-CC scaling for present day climate. This indicates that beyond purely thermodynamic processes, also the dynamic component within convective clouds contributes to the intensification and has to be evaluated separately. However, it has to be mentioned that some studies have found a scaling that is close to the CC rate, like Ban et al. (2015) who argue that the super-CC scaling might be an artifact that results from the statistical methods applied to determine the scaling rate, e.g. the imposition of thresholds for wet days in the data analysis.

Analyses of climate change projections have indicated that while the thermodynamic contribution to the intensification of extreme precipitation is expected to be relatively homogeneous globally, there may be strong regional differences in the dynamic contribution due to changes in circulation patterns (Emori and Brown, 2005; Pfahl et al., 2017; Norris et al., 2019). CRMs should therefore also be able to simulate the time evolution of convective precipitation events and their interaction and organization among each other in a realistic way, to correctly represent their sensitivity to 2m air temperature. Ban et al. (2014) have analyzed the temperature scaling of a decade-long simulation with 2.2 km grid spacing over Switzerland, and found a good agreement with observations. Kendon et al. (2014) have found an intensification of hourly rainfall over Britain under a climate change scenario with a 1.5 km model. However, a correct representation of the temperature scaling of heavy rainfall becomes increasingly difficult with decreasing model resolution. Rasp et al. (2018) have shown that in principle subgrid cloud organization has to be included into stochastic cloud parametrizations. These parametrizations are particularly relevant at the above-mentioned convection permitting scale, and at present assume a random cloud distribution within model grid cells.

Some studies that have investigated the sensitivity of convection to resolution, distinguish between *bulk convergence* and *structural convergence* (Langhans et al., 2012; Panosetti et al., 2019): while the former is concerned with large-scale mean properties, the latter refers to an analysis of cloud sizes, cloud shapes, and convective organization. Our study mainly addresses structural convergence. To analyze the properties of convection and convective organization in model output and gridded observations like radar or satellite data, object-oriented methods are increasingly applied. Besides simple mean values and percentiles of precipitation intensities, they provide information on the spatial distribution of sizes and shapes of precipitation objects. Furthermore, several indices that are based on these methods have been developed over the recent years, that are capable of quantifying the degree of organization of the convection cells in space (Senf et al., 2018; Pscheidt et al., 2019). Using a combination of several convective organization indices that we also apply in the present study, Pscheidt et al. (2019) have shown that convective precipitation cores and cloud-tops are organized most of the time over Germany.

However, the shortcoming of these methods is that they provide only information on the spatial distribution of convection objects, but not on their temporal evolution. Tracking methods are able to additionally capture the life cycles of the objects, and their interaction among each other. Several tracking methods for convective storms have been developed in the past, and although they are based on similar ideas, they are specialized for different purposes, such as nowcasting thunderstorms (Dixon and Wiener, 1993; Hering et al., 2005; Kober and Tafferner, 2009; Wapler, 2017), studying the cloud life cycle statistics in shallow (Heus and Seifert, 2013; Heiblum et al., 2016), and deep convection (Lochbihler et al., 2017; Moseley et al., 2019), or even larger structures like mesoscale convective systems (Fiolleau and Roca, 2013).

In this study, we apply the tracking method of Moseley et al. (2019), which provides statistical information on the interaction of convective precipitation objects among each other in terms of merging and splitting. We analyze convective diurnal cycles simulated by the ICON-LEM with grid spacings in the sub-kilometer range, and assess the impact of horizontal resolution, and daily mean temperatures, on the simulated convection. This article is organized as follows: in section 2 we describe the ICON-LEM setup, the radar dataset that is used for evaluation, and the object-oriented analysis methods. In section 3, we compare the simulation results of three different model resolutions between 625 m and 156 m grid spacing, and in section 4 we analyze a continuous 36 day long simulation period with 625 m grid spacing. We discuss results in section 5, and conclude in section 6.

## 2 Data and methods

### 2.1 Model configuration

The simulations are performed with the unified modeling framework ICON which was run with the LES physics package, in the following termed ICON-LEM ("ICON Large Eddy Model") (Dipankar et al., 2015). ICON is a non-hydrostatic new-generation model tailored to perform atmospheric simulations in different setups ranging from global climate reconstructions to limited-area nested configurations and idealized configurations. Different physics packages needed to parametrize sub-scale variability are adopted depending on the setup considered. ICON is used at the German Weather Service (DWD) since 2015 to produce operational forecasts and has been successfully adopted as tool to improve our understanding of moist convection in

many areas of the world (e.g. Klocke et al. (2017)).

In our work, ICON-LEM is used in a limited area configuration to perform convection-explicit simulations over Germany. The model configuration follows very closely the description given in Heinze et al. (2017), to which the reader is referred for further details on the parametrizations employed. We only emphasize that turbulence is parametrized using a Smagorinsky model (Dipankar et al., 2015) (thus, subgrid turbulence is treated as isotropic), the land surface is described using the TERRA-ML model (Schrodin and Heise, 2002), the surface layer is treated with a drag-law formulation following Louis (1979), a simple all-or-nothing cloud scheme is used, and cumulus convection, as well as gravity waves (orographic and non-orographic) are not parametrized.

At the boundaries, ICON is forced by operational hourly analysis data by the previous operational NWP model COSMO-DE by the DWD, run with ca. 2.8 km grid spacing. The model output is interpolated to the ICON model grid with 625 m grid spacing, on which the model simulations are performed. Dynamical downscaling in a one-way nesting approach is applied on 3 of the model days, in a first step to 312 m, and in a second step to 156 m grid spacing (Heinze et al., 2017). In this case, boundary conditions for each one of the two inner domains are taken from the relative outer domain (see Fig. 1).

We note that we restrict the evaluation of the ICON-LEM simulations to daytime between 6 and 21 UTC, since it is known that the nocturnal boundary layer is not sufficiently resolved at LES resolutions of 100 m and coarser, which may introduce unknown biases in cloud cover at night van Stratum and Stevens (2015). Therefore, the figures showing our results are also restricted to this period.

## 2.2 Simulation period

We chose a period of 36 continuous days, beginning on May 26, 2016 and ending on June 20, 2016. This period includes an exceptional sequence of severe weather events producing heavy convective precipitation, 10 tornadoes and hail, which caused damages running into the billions of Euros (Piper et al., 2016). The strongest events were concentrated between 26 and 29 May 2016 mostly over Southern Germany, while during the first days of June a $\Omega$-blocking pattern over Europe prevented the typical westerly flow from reaching central Europe and enhanced local instability caused by diurnal surface heating and nocturnal cooling.

To reduce computational costs, the entire 36-days period is simulated only on the outermost nest (domain 1) with 625 m grid spacing. The simulation is initialized on May 26, 2016 at 00:00 UTC and continuously run through June, 31 2016 00 UTC using only the forcing from the boundary conditions provided by hourly analysis of the COSMO-DE data at the lateral boundaries of the outer domain. Local features, such as individual clouds or thunderstorms, are mostly the results of local forcing and thus may look different from the observed ones, which is partially due to the inherent unpredictability of convection. Three days among this period are simulated with the additional nests with 312 m and 156 m grid spacing (a more detailed description of the large-scale situation in this period over Germany is given by Rasp et al. (2018), who analyzed the period between May 26 and June 9, 2016, in their study):

– **May 29, 2016**, was dominated mainly by wind from the South East, with relatively widespread high level clouds that grew larger throughout the afternoon, and strong convection over the largest part of the domain. At night, a mesoscale convective system developed that covered most of southern Germany.

– **June 3, 2016**, was characterized by moderate Easterly wind in the Northern half of the domain with mainly clear sky in the morning and broken convective cloudiness in the afternoon. The Southern part of the domain was dominated by strong convective rainfall, beginning around noon.

– **June 6, 2016**, was characterized by weak Easterly winds, and a distinct diurnal cycle of convection with mainly clear sky in the morning, and convective cloudiness with a maximum in the afternoon over the largest part of the domain, associated by increasing high level cloudiness caused by stratiform outflow.

In all simulations, the state of the atmosphere and the soil has been initialized at 0 UTC with COSMO-DE data. The first 6 simulation hours are used as spin-up for the atmosphere, and are removed from the analysis. For the high-resolution 3-domain simulations, all 3 nests are initiated at the same time.

## 2.3 Preparation of model and radar data

We use the RADOLAN RY C-Band weather radar composite provided by the German Weather Service (Bartels et al., 2004). This data product contains precipitation intensities derived from radar reflectivities on a grid of approximately $1 \times 1$ km$^2$. We apply a conservative remapping to interpolate all model and radar data to a common lat-lon grid. This implies that we also evaluate the model data on the three nests with 625 m, 312 m, and 156 m model grid spacing on the same target grid after interpolation. The main reason for interpolating the model data is that the original ICON-LEM output is given on an unstructured triangular grid which is difficult to handle for our post-processing tools. The second reason is that we prefer to compare the data of the three different model resolutions, and the radar data, on the same grid, to reach a fair comparison. We chose a $1 \times 1$ km$^2$ lat-lon grid, since this is roughly the resolution of the radar data. Further, it is only slightly coarser than the resolution of the coarse ICON-LEM resolution with 600 m grid spacing of the triangle edges. However, as the *effective* resolution of the ICON-LEM data is larger than the grid spacing, we can assume that there is no loss in resolution at least for the 600 m simulation. A similar regridding has also been used for other studies which also analysed ICON-LEM output (Heinze et al., 2017; Pscheidt et al., 2019).

As the radar data contain areas of missing values that vary in time when instruments were switched on and off, we mask out these areas also in the model data, to have a one to one comparison. In section 3, where we compare results of all three nests, we restrict the domain to the innermost nest with 156 m grid spacing as shown in Fig. 1. Elsewhere, where we analyze only the outer domain with 625 m grid spacing, we include the full domain size.

The temporal output interval of the model data is 2 min, while the radar data are available with a 5 min interval. Therefore, the modeled precipitation intensities have been linearly time-interpolated to a 5 min interval.

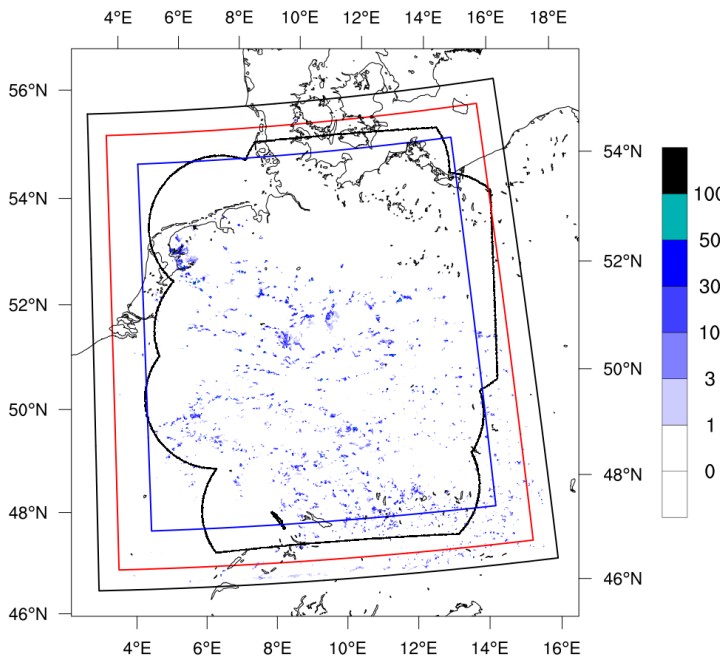

**Figure 1.** Simulation domain. The black frame shows the extent of the outer domain 1 with 625 m grid spacing, the red and blue frames show the nested domains 2 and 3 with 312 m, and 156 m grid spacing, respectively. The black contour shows the maximum extent of the RADOLAN dataset. Color shading shows the surface precipitation field on June 3, 2016, at 14:00 UTC, simulated on the outer domain with 625 m grid spacing, given in [mm h$^{-1}$].

## 2.4   Indices of convective organization

To investigate whether convective clouds tend to organize in space, we follow the approach used in Pscheidt et al. (2019): first, we detect signatures of convection in radar and model rain rates by applying a segmentation algorithm with a split-and-merge approach (Senf et al., 2018) with a threshold of 1 mm h$^{-1}$. In a second step, we compute commonly used organization indices for the radar observations and the simulation output. The organization indices are based on the characteristics of the 2D objects obtained from the segmentation algorithm. We employ three organization indices, namely the Simple Convective Aggregation Index (SCAI, (Tobin et al., 2012)), and the Convective Organization Potential (COP, (White et al., 2018)), which are both based on all-neighbors distances, and the $I_{\mathrm{org}}$ index (Tompkins and Semie, 2017), which uses a nearest neighbor (NN) distance approach. SCAI is defined as

$$\mathrm{SCAI} = \frac{ND_0}{N_{max}L}1000,$$  (1)

where N is the number of objects in the domain, $D_0$ is the geometric mean distance of the centroids between all possible pairs of objects, $N_{\max}$ is the possible maximum number of objects that can exist in the domain, and L is the characteristic domain size. In this study, $N_{\max}$ is the total number of grid boxes in the domain, and L is the Southwest-Northeast distance in the domain. The degree of organization increases as the SCAI decreases.

5    COP considers the interaction potential between two objects $V(i,j) = (\sqrt{A(i)} + \sqrt{A(j)})/(d(i,j)\sqrt{\pi})$, where $A(i)$ is the area of object $i$ and $d(i,j)$ is the Euclidean distance between the centroids of the objects $i$ and $j$. COP is defined as

$$\text{COP} = \frac{\sum_{i=1}^{N}\sum_{j=i+1}^{N} V(i,j)}{\frac{1}{2}N(N-1)}. \tag{2}$$

The degree of organization increases as COP increases.

Unlike SCAI and COP, which mainly quantify the degree of clustering, the NN-based organization index $I_{\text{org}}$ (Tompkins and Semie, 2017) is able to distinguish between three types of spatial distribution: clustered, regular, and random. In this approach, we treat objects as discs (similar to Nair et al., 1998), and compute the cumulative distribution function of the NN edge-to-edge distances (NNCDF) and compare it to the NNCDF of theoretical randomly distributed objects over the same domain. The theoretical NNCDF is approximated by bootstrapping, in which a random number of objects with the observed size distribution is randomly placed over the domain (Weger et al., 1992; Nair et al., 1998). We perform 100 simulations and compare the observed NNCDF to the 100 theoretical NNCDFs in a graph. $I_{\text{org}}$ is defined as the area below such a comparison curve (for more details see e.g. Pscheidt et al., 2019; Tompkins and Semie, 2017). From the 100 computed $I_{\text{org}}$ indices we select the 2.5th and 97.5th percentiles to identify the spatial distribution. The objects are organized in clusters when the 2.5th percentile is greater than 0.5, whereas they present a regular distribution in space when the 97.5th percentile is lower than 0.5. Otherwise, the scenario can not be differentiated from randomness.

In addition to the degree of convective organization, we also investigate the shape of the objects with the index $I_{\text{shape}}$ defined as

$$I_{\text{shape}} = \frac{1}{N}\sum_{i=1}^{N} s(i), \tag{3}$$

where $s(i) = P_{\text{eq}}(i)/P(i)$ is the shape ratio, P(i) is the actual perimeter and $P_{\text{eq}}(i) = \sqrt{4\pi A(i)}$ is the perimeter of an equivalent, area-equal disc of the object i. The perimeter P(i) is computed as the contour line through the centers of the border grid boxes of the objects (Benkrid and Crookes, Online; accessed 2017; van der Walt et al., 2014). $I_{\text{shape}}$ ranges between 0 and 1 and indicates the predominant presence of linear shapes for the former and circular shapes for the latter. $I_{\text{shape}}$ close to 0.5 indicates predominance of elliptical shapes.

## 2.5    Rain cell tracking

We apply the "Iterative raincell tracking" (IRT) algorithm to track life cycles of convective precipitation events in space and time (Moseley et al., 2019). In a first step, precipitation objects are detected for each time step individually. They are defined

as connected areas over a given threshold chosen as 1 mm h$^{-1}$ surface precipitation intensity. This threshold has proven to generate reasonable results, and is in the order of the resolution threshold of the weather radar. For each object, the area, and the mean surface precipitation intensity averaged over this area is recorded. The algorithm checks for overlaps of each object with objects in the previous, and the subsequent time step, and records the concerning object identifiers. If an object overlaps with more than one object at the previous or subsequent time step, the two largest ones are recorded, others are ignored.

It sometimes happens that objects of subsequent time steps do not overlap although they belong to the same track, since they are advected by mean background flow, especially if the time step is relatively large and the objects are small. To correct this artifact, in a second step a mean background advection field is diagnosed and the procedure is repeated by taking into account the displacements of the objects due to the advection field while checking for overlaps. This step has to be iterated until the object identification result converges.

In a third step, overlapping objects are combined to tracks. A fraction of the tracks have distinct life cycles, and do not merge with others, nor split up into fragments. They are initiated as new emerging precipitation events and eventually vanish when surface precipitation ceases. We call these tracks *solitary*. Tracks that experience merging and splitting are recorded separately. We call these tracks *interacting*. A parameter, the so-called *termination sensitivity* $\Theta$ that takes values between 0 and 1, provides a criterion whether a merging or splitting event is recorded, or ignored. If $\Theta = 0$, then *every* merging and splitting event will lead to a termination of all involved tracks, and will be recorded as a tracks that interacts with its neighbours. In the other extreme $\Theta = 1$, the largest object that experiences a merging or splitting event will always be continued and regarded as solitary, while the smaller involved tracks will be terminated and not be regarded as non-solitary. If $\Theta$ takes intermediate values, all participating tracks will only terminate, when they are of comparable size, otherwise the largest one will regarded as solitary, and the smaller one as interacting. For our analysis, we choose an intermediate value of $\Theta = 0.5$.

## 3 Impact of resolution

### 3.1 Domain mean precipitation and size distribution

To analyze the impact of resolution on the simulated life cycles of convection, we make use of the three days which have been simulated on three nests with 625 m (DOM01), 312 m (DOM02), and 156 m (DOM03) grid spacing. Fig. 2 shows the time series of the daily mean precipitation for each day for all three domains next to the radar data, averaged over all areas where radar data are available. While on May 29 the simulated precipitation amount on all three domains is very close to each other and strongly mismatches the radar data, on the other two days in June 3 and 6 the time evolution of mean precipitation differs more strongly for the different resolutions. On the latter two days, the match with RADOLAN is better for higher resolutions: the peak precipitation on the 625 m domain is larger, and is reached earlier than for the 312 m and 156 m nests. Especially on June 3, both the magnitude and the timing of the precipitation peak is closer to the radar data for the domains with higher resolutions than for the 625 m domain. On both June 3 and 6, the strong increase in precipitation around 10:00 UTC is steeper than in the radar data for 625 m, while the slope matches the radar data best for 156 m. However, on June 6 the decline of precipitation intensity in the late afternoon and evening hours appears too late. We note that although the later onset in the

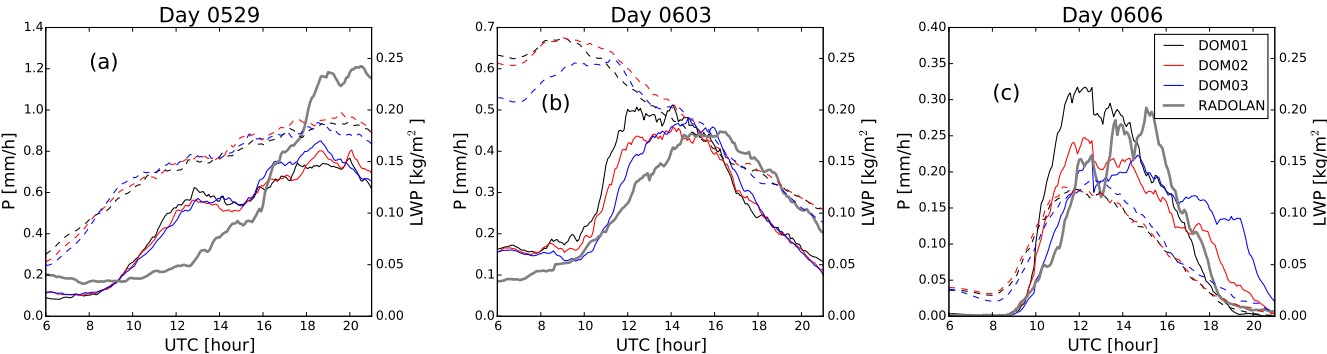

**Figure 2.** Time series of the mean precipitation intensity $P$ (solid lines, left axes) and liquid water path LWP (dashed lines, right axes), for the three days May 29 (a), June 3 (b), and June 6 (c), on all three domains with 625 m (DOM01), 312 m (DOM02), and 156 m (DOM03) grid spacing. The gray thick line shows the RADOLAN derived precipitation intensity. Averaging was done over all grid boxes where radar data are available.

simulations with higher resolution appears to be consistent on June 3 and June 6, we cannot rule out that some of the other differences may be due to internal variability, like individual large storms. Simulated cloud water follows the total precipitation intensity closely on the days May 29 and June 6, while the high values of LWP in the morning hours on June 3 indicate non-precipitating cloudiness, which was found mainly in the southern part of the domain.

5    Rain cell size distributions for all three nests and RADOLAN and shown in Fig. 3, including the rain cell objects of all 3-domain days. Compared to the 625 m nest, the RADOLAN data show a larger fraction of large objects, but fewer small objects that can be attributed to isolated cells. However, the *total* number of large clouds in the radar data is not much different from the simulations. For the higher resolved nests, the fraction of small objects is closer to radar. This picture is consistent if the size distribution is plotted for each of the days individually (not shown).

## 10    **3.2    Convective organization indices**

A general convergence of the higher resolution nests to the RADOLAN data is not only found in the diurnal cycles of mean precipitation and the cell size distribution, but also in the organization indices that we have calculated on the three domains and the RADOLAN data, especially in SCAI and $I_{\text{shape}}$ (Fig. 4). In general, SCAI tends to follow the mean precipitation intensity, rather than the mean amount of cloud water (Fig. 2). The analysis of SCAI reveals that on May 29 the radar objects are more

15  clustered than the simulated ones (Fig. 4a), however, the finest nest is closest to RADOLAN. The 156 m nest also shows the best performance during June 6, when the degree of organization of observed objects is very well represented at 156 m (Fig. 4c). The situation is, however, different for June 3 (Fig. 4b). Before 12:00 UTC the finer nests represent best the degree of organization, whereas from 12:00 UTC until 18:00 UTC, the coarsest nest is in better agreement with radar. On all three days, SCAI shows a clear increase in the degree of clustering with the nest's resolution, which is due to the decrease in the number of

20  small objects as the grid spacing increases (see the size distribution in Fig. 3). Although the size distribution does not provide

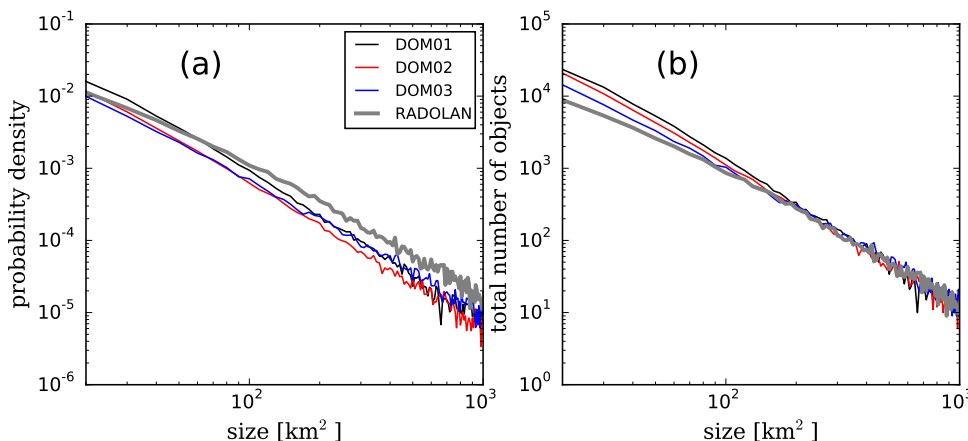

**Figure 3.** (a) Normalized probability density function (PDF) of rain cell size distributions, on all three domains DOM01, DOM02, DOM03, and the RADOLAN data. (b) Same as (a), but with total number of cells on vertical axis (in bins of width 10 km$^2$. The PDF includes all rain cells between 6 and 21 UTC on all three days (May 29, June 3, June 6).

any direct information on the shape of objects, the smaller value of $I_{\mathrm{shape}}$ in the radar data is consistent with the larger fraction of large objects, since large objects are more likely to deviate strongly from the circular shape (Fig. 4j–l).

The COP index indicates more clustering of the radar objects than in the simulations in the course of the days, especially on May 29 and June 3, due to the smaller sizes of the simulated objects (Fig. 4d,e; note also the size distributions in 3). A clustered

distribution is also reinforced by $I_{\mathrm{org}}$ (Fig. 4g–i), indicating convective organization throughout the day with a slight decrease in the degree of clustering in the afternoon in agreement with SCAI and COP. The simulations represent $I_{\mathrm{org}}$ in all three grid spacings well and significant differences among the three grid spacings are not found. In contrast, the shape of the objects are best represented for the 156 m nest for the days May 29 and June 3, with decreasing performance for the coarser nests (Fig. 4j,k).

For June 6, the diurnal cycle of the COP and $I_{\mathrm{org}}$ show a different behavior than on the other two days: COP is in good agreement with radar between 09:00 UTC and 17:00 UTC for all three grid spacings (Fig. 4f). In the evening, however, the simulations with the finest nests reveal larger object sizes (not shown) than observed in radar leading to an overestimation of the degree of clustering. Besides, no objects are detected in the 625 m nest after 19:00 UTC. The increased oscillation in the degree of clustering after 20:00 UTC seen in COP is reflected in $I_{\mathrm{org}}$, and indicates spatial distributions varying between

clustering and random distribution (Fig. 4i). Regarding the object's shapes, the coarsest nest shows the best performance for this day, though (Fig. 4l).

An interesting observation is that SCAI differs more strongly between the days, while for the other indices, the differences among the simulation nests and the radar data are of the same order as the differences between different days. The reason could be that SCAI follows closely the total number of rain cells which varies strongly between days, while the other indices are

rather linked to the size distribution which is similar on all days.

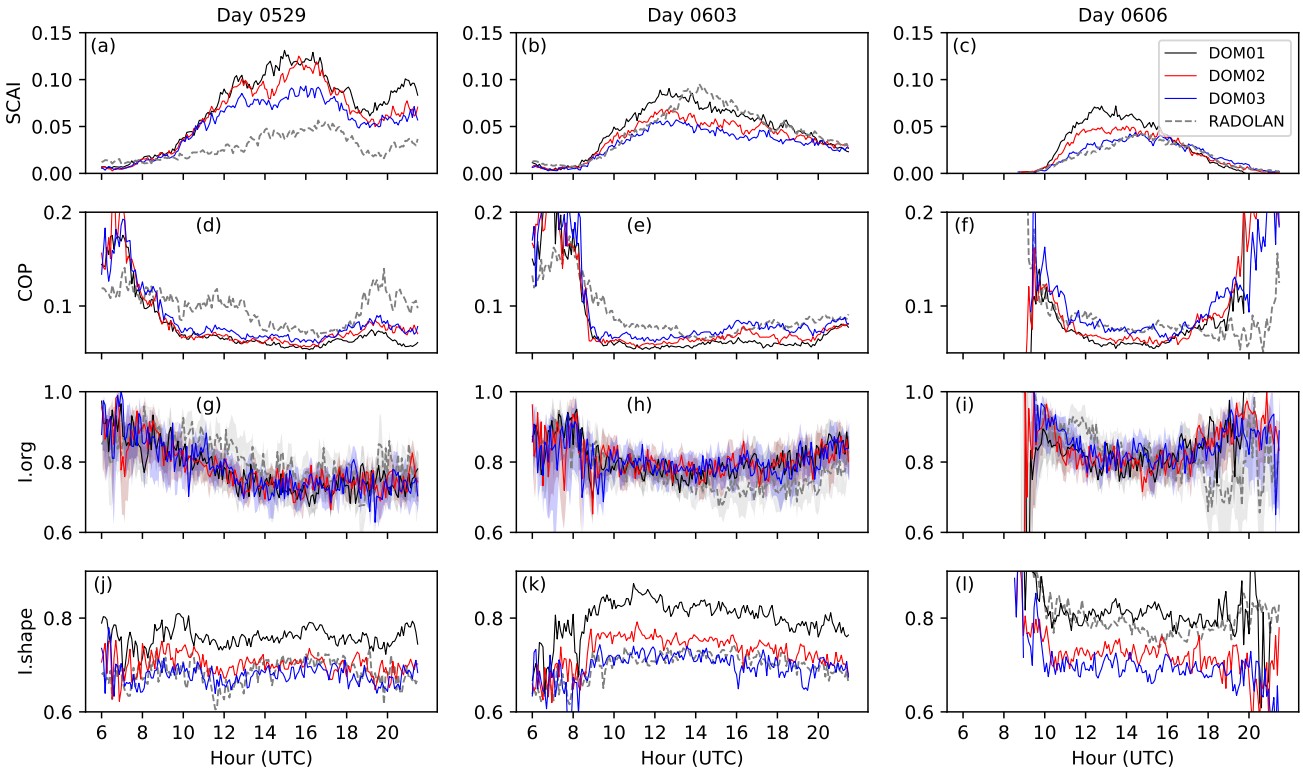

**Figure 4.** Convective organization indices SCAI (a-c), COP (d-f), median $I_{\mathrm{org}}$ (g-i) and $I_{\mathrm{shape}}$ (j-l) for the days May 29, June 3 and June 6, 2016, for the three nests with 625 m (DOM01), 312 m (DOM02), and 156 m (DOM03) grid spacing, and the RADOLAN data. Averaging was done over all grid boxes within the 156 m nest where radar data are available.

## 3.3 Track statistics

We apply the tracking algorithm on the precipitation cells of model and RADOLAN data (note that all data are evaluated on the domain of the innermost nest, and on the same grid), and build a single sample containing all tracks of the three days. In total, the algorithm detects 141682 tracks for DOM01, 160042 tracks for DOM02, and 124820 for DOM03, showing no clear trend with resolution. For the radar data, a smaller number of 67657 tracks is detected. We perform a separate analysis for solitary tracks (i.e. tracks that do not merge or split), for tracks that involve only merging (i.e. tracks that either merge into others or are initiated by merging of other tracks, but that do not involve splitting), tracks that involve only splitting (i.e. tracks that split up, or tracks that are initiated as a fragment of a splitting event), and tracks that involve both merging and splitting (i.e. tracks that either are initiated as a merging event, and split up later, or that are initiated as a fragment and later merge again with other tracks), see Table 1. Although less than 10% of the total rainfall is generated by solitary tracks (excluding drizzle below the threshold of 1 mm h$^{-1}$, and tracks that touch the boundaries), there is a strong variation of the contribution of solitary tracks

**Table 1.** Ratio of the number of tracks of given track types (solitary, tracks that involve only merging, tracks that involve only splitting, and tracks involving both merging and splitting), and the total amount of rainfall that they contribute, relative to the total number, and rainfall amount, respectively, of all tracks. Note that tracks that touch the domain boundaries are removed from the analysis. Fractions (in [%]), including all three 3-domain days, are given for all three domains with 625 m (DOM01), 312 m (DOM02), and 156 m (DOM03) grid spacing, and for the RADOLAN composite.

| Ratio (number; amount) [%] | DOM01 | DOM02 | DOM03 | RADOLAN |
|---|---|---|---|---|
| Solitary tracks | 34.0; 9.4 | 32.7; 7.1 | 32.3; 4.2 | 31.5; 6.7 |
| Involving only merging | 25.4; 36.7 | 26.3; 27.7 | 26.7; 28.2 | 26.5; 26.7 |
| Involving only splitting | 28.5; 24.1 | 28.7; 25.4 | 27.7; 20.4 | 27.5; 34.2 |
| Both merging and splitting | 12.1; 29.7 | 12.3; 39.8 | 13.3; 47.2 | 14.5; 32.4 |

to the total rainfall, namely 9.4%, 7.1%, 4.2%, indicating the tendency toward more organization with increasing resolution. For comparison, for RADOLAN we find a fraction of 6.7%, which is between the model results of the 312 m and 156 m nest. The ratio of the *number* of tracks belonging to all track types is very similar for all nests and matches well with RADOLAN, but there are differences in the contribution to total rainfall among these types. There is a clear increase with resolution from 29.5% (for DOM01) to 47.2% (DOM03) for the type that experiences both merging and splitting. As this track type can be regarded as the one that experiences the strongest interaction with neighbouring tracks, the high rainfall ratio falling onto this track type at the 156 m nest indicates a stronger impact of convective organization. However, for RADOLAN, this ratio is only 32.4% which is close to the coarse resolution result.

Even though solitary tracks contribute to less than 10% of the total precipitation, they are most suited for an analysis of the time evolution of convective rainfall events. Therefore, we have a closer look at the performance of the model to simulate solitary track life cycles. Mean life cycle composites of the three 3-domain days, comparing model and RADOLAN tracks and conditioned on short (20–40 min), intermediate (50–70 min), and long (80–100 min) track durations, are shown in Fig. 5. The curves show that generally the mean peak intensities get lower for higher resolutions, while the largest jump is visible between 312 m and 156 m grid spacing (Fig. 5a–c). The match with RADOLAN intensities is best for the 156 m nest. The track sizes do not show an improvement with increasing resolution compared to the radar data: sizes are smaller in the model data than in RADOLAN, except for short duration tracks in the 625 m domain. In contrast to intensity, track maximum extents of the 625 m domain show a better match with RADOLAN, while the sizes of tracks of the 312 m and 156 m nests are clearly smaller (Fig. 5d–f). The rate of total precipitation produced by the solitary rainfall events (i.e. the spatial integral of precipitation intensity integrated over the object area shown in Fig. 5g–i), however, shows that for intermediate and long duration tracks simulated with 625 m grid spacing, the too large intensities are compensated by the too small intensities, resulting in a good match with RADOLAN, while rates are clearly too small for the finer nests. Only for the short duration tracks, the precipitation rate of the 156 m nest agrees with RADOLAN, while the coarse resolution produces too much precipitation.

We further visualize the statistics of the solitary track peak intensities, the maximal effective radii of the objects (where the effective radius is given as $r_i = \sqrt{A_i/\pi}$ with $A_i$ being the area of object $i$), and the total precipitation amount produced by the

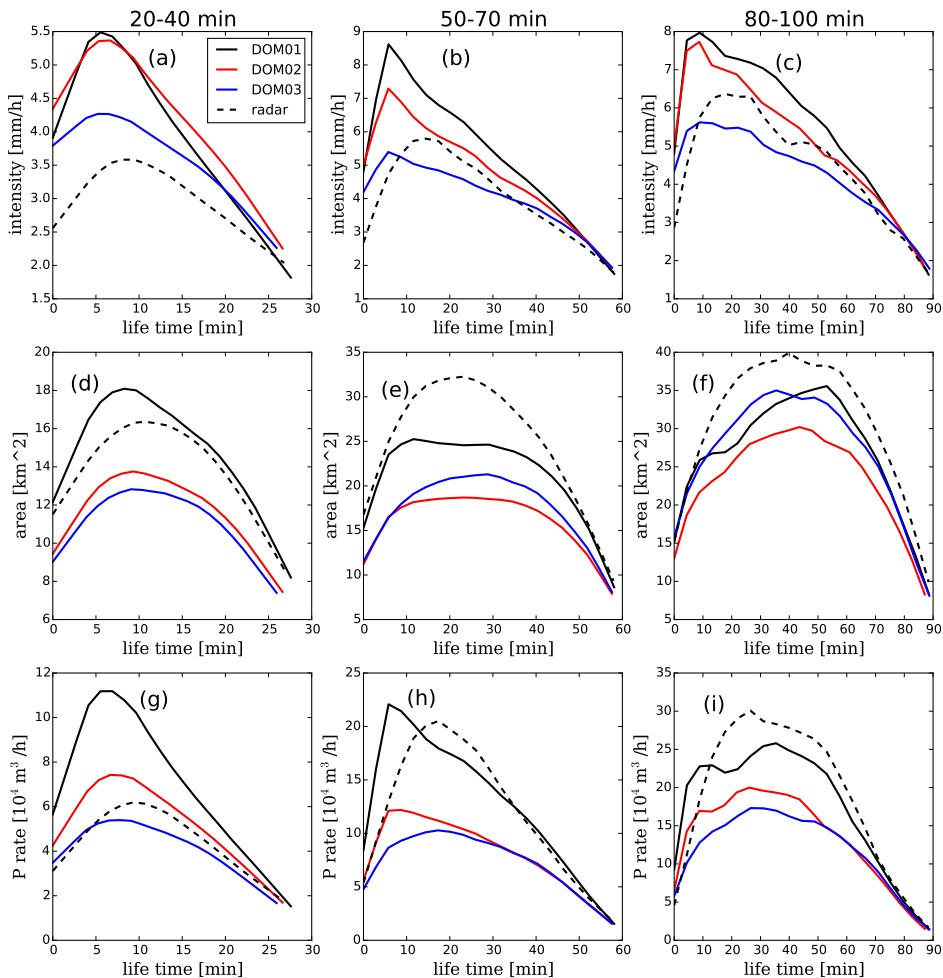

**Figure 5.** Life cycles of track composites (including the days May 29, June 3, and June 6) for solitary tracks of different track duration for model results on three domains with 625 m (DOM01), 312 m (DOM02), and 156 m (DOM03) grid spacings, and for radar results. Curves show mean track life cycles of area-mean precipitation intensity (a–c), area of precipitation objects (d–f), and rate of total precipitation (that is the areal integral of local precipitation intensity over the object extent) (g–i), conditioned on tracks with durations between 20–40 min (a,d,g), between 50–70 min (b,e,h), and between 80–100 min (c,f,i).

tracks (given as the spatial integral over the area, and the temporal integral along the track duration, of the local intensity), in the box-and-whisker plots in Fig. 6. The solid curves in Fig. 6a show that in total there are more solitary tracks found in the model data than in RADOLAN, but for longer durations, curves for RADOLAN and the 156 m nest converge. The decreased number of longer lasting solitary tracks reflects the stronger organization at the high resolution domain, since stronger convective events are more likely to interact with neighbouring tracks. As already indicated by the life cycles in Fig. 5, we see that the median of peak intensities is lowest for the finest resolution and shows a good match with RADOLAN, while peak intensities reach higher values for the 625 m domain. However, the spread in peak intensities is much higher for the RADOLAN data for longer duration tracks, while it is lowest for the 156 m nest, a feature that is not visible in the mean life cycles in Fig. 5. Further, Fig. 6b confirms that RADOLAN track maximum sizes are best matched with the coarse 625 m domain, while sizes are smaller at higher resolutions. The spread of the maximum size distribution is relatively narrow compared to intensities, and is similar for all resolutions and for the RADOLAN data. Not surprisingly, the resulting total amount of precipitation produced by the tracks (Fig. 6c) strongly increases with track duration. For tracks longer than 1 hour, the spread of the inner quartiles between model data and RADOLAN matches best for the 625 m domain, while the median matches better with the finer nests, although they show a clearly smaller spread.

To briefly summarize this section, both the convective organization indices and the rain cell tracking show that for the higher resolution nests there is a stronger tendency of convection to organize, which generally provides a better match with RADOLAN data. Further, convective precipitation increases more rapidly at the onset of convection at 625 m grid spacing, compared to the finer resolutions, and the RADOLAN data. This can be seen both in the diurnal cycle of mean precipitation, and the life cycle composits of the solitary tracks. Although three model resolutions are insufficient to clearly identify bulk convergence and structural convergence, these results show an improved simulation of convection at the 100 m scale with ICON-LEM.

## 4   Analysis of the continuous 36 days period with 625 m grid spacing

### 4.1   Mean diurnal cycles

In the previous section we argued that the ICON-LEM setup with 625 m grid spacing is sufficient to reasonably simulate typical convective summer days over Germany, although there may still be room for added value at even higher resolutions. We now discuss the continuous simulation period from May 26 until June 20, 2016, simulated with 625 m grid spacing. The simulated domain mean precipitation with the RADOLAN data for the full period is shown in Fig. 7. On some of the days we see an underestimation of simulated rainfall compared to RADOLAN, like on May 30, June 12, June 16, and in the 3-day period between June 23–25. However, there are few days where the precipitation intensity is slightly overestimated, like on June 19 and June 26. Another mismatch between model and radar data is that daily peak intensities tend to be reached 1–3 hours earlier in the model simulation compared to RADOLAN. This is particularly visible in the 6-day period June 3–8. This feature can be explained by the observation discussed in the section 3, where we argued that convection is triggered too fast in the 625

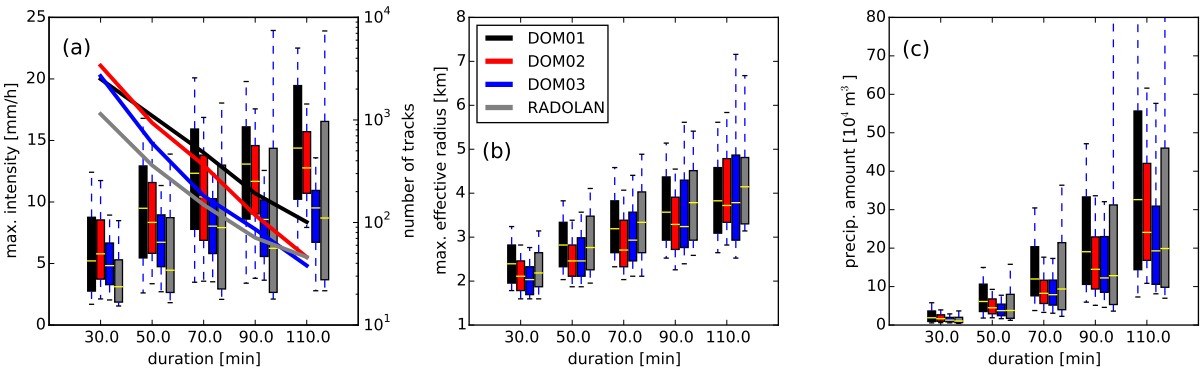

**Figure 6.** Box-and-whisker plots showing the statistics of solitary tracks, including the days May 29, June 3, and June 6 on all three domains with 625 m (DOM01), 312 m (DOM02), and 156 m (DOM03) grid spacing, and for the radar data. Values of track maximum intensity (a), track maximum effective radius (b), and total precipitation amount produced by the individual tracks (c), are conditioned on track duration ranging between 20 and 120 min, in 5 bins of 20 min width. Boxes indicate the 25th and 75th percentiles and the median (yellow bar), whiskers indicate the 10th and 90th percentiles. The number of tracks in each bin is indicated by the solid lines in panel (a) (note the logarithmic axis on the right).

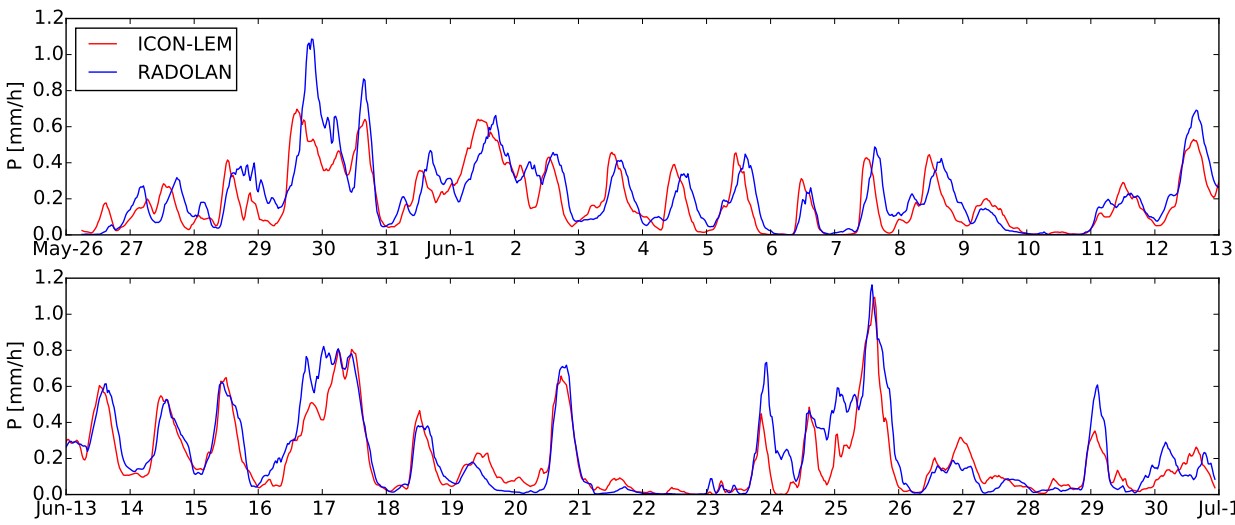

**Figure 7.** Time series of the mean precipitation intensity $P$ for the 625 m grid spacing ICON-LEM simulation, and the RADOLAN derived precipitation intensity, for the full 36-days period from 26th May 2016 until 30th June 2016. Note that the time series was broken into the upper and the lower panel. Averaging was done over all grid boxes where radar data are available.

m LES simulation. We note that in addition to these systematic differences, some of the differences between model and radar data could also be traced back to the uncertainty in boundary conditions from the COSMO forecast data.

To confirm that the simulated 36-day convective period is long enough to show the intensification of convection with higher temperatures as discussed in the introduction, and that it is also simulated with ICON-LEM and 625 m grid spacing, we perform a separate analysis for selected cool and warm days. We calculate the domain mean temperature from the original COSMO-DE forcing data, and average over the time between 8:00 UTC and 20:00 UTC when daytime convection is expected. We hereby use the original COSMO-DE analysis data that provided the forcing, as we expect them to be closer to the actual temperatures than the temperatures simulated by ICON-LEM. We classify days below 16 °C daytime mean 2 m temperature as cool, and between 19 °C and 21 °C as warm. The two exceptionally warm days June 23 and June 24 with mean temperatures of 26.0 °C and 24.1 °C, respectively, are not included in the ensemble of warm days. Further, the day June 22 was removed from the classification due to the very low precipitation amount (otherwise it should have been classified as a warm day). An overview of the classified days can be seen in Table A1. In total, out of the 36 days of the simulation, we classify 6 days as cool, and 6 days as warm.

Mean diurnal cycles of several domain averaged quantities, including all 36 days, and conditioned on cool and warm days, are shown in Fig. 8. As already mentioned, the peak in mean precipitation (Fig. 8a) appears earlier in the model than in the RADOLAN data, and it is higher for the cold days than for the total mean of all days. For warm days, the peak is also slightly larger than for the total mean, although there is less precipitation in the afternoon hours after 15:00 UTC. The simulation period is too short to significantly state if there is any direct correlation between the total amount of precipitation and the daily mean temperature. However, there is clear temperature dependence of the 99th percentile of precipitation intensity (Fig. 8b): in consistency with the CC argument mentioned in the introduction, there is less (more) water vapor available in the atmosphere on cool (warm) days than on average (Fig. 8c), associated with lower (higher) extreme rainfall intensities. However, the differences in the 99th percentile of precipitation are more pronounced in the RADOLAN data, suggesting that the sensitivity of heavy rainfall to temperature is underestimated by the model. Further, we see that cool (warm) days are associated with lower (higher) surface fluxes (Fig. 8d–f). As a sensitivity test, we randomly chose 3 out of the 6 warm days, and 3 out of the 6 cool days, and reproduced the plot in Fig. 8b with these days (not shown). Repeating this procedure 4 times confirmed that peak intensities of the 99the percentiles are stronger (weaker) for warm (cool) days in both radar and model data, and second, that the difference between warm and cool days is weaker in the model than in the radar data.

## 4.2 Diurnal cycles of convective organization indices

We calculate mean diurnal cycles of the convective organization indices SCAI, COP, $I_{\mathrm{org}}$, and $I_{\mathrm{shape}}$, for model and RADOLAN data of all 36 days, and conditioned on cool and warm days (Fig. 9). SCAI, COP and $I_{\mathrm{org}}$ indicate more organization in the morning and evening, when the objects present also a more elliptical shape (Fig. 9d). During the afternoon, when the convective activity is more intense, there is a decrease in the degree of organization, with the shape of the objects tending towards a more circular one. ICON reproduces the diurnal cycle of $I_{\mathrm{org}}$ very well (Fig 9c). Although the variability of SCAI, COP and $I_{\mathrm{shape}}$ are captured by the model at 625 m grid spacing, it underestimates the degree of organization revealed by RADOLAN (Figs. 9a-b) and produces more rounded objects than the radar observations (Fig. 9d) especially in the afternoon, as was discussed in section 3.2.

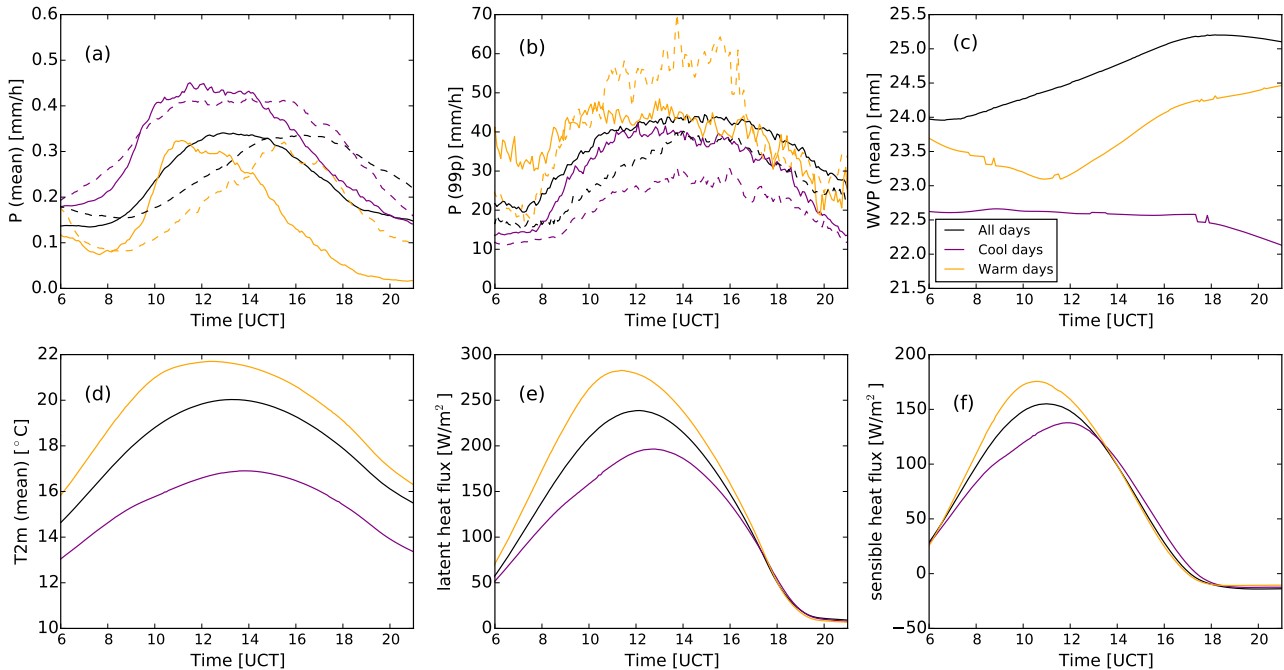

**Figure 8.** Diurnal cycles of mean precipitation intensity (a), 99th percentile of precipitation intensity (b), water vapor path (c), air temperature at 2 m (d), surface latent heat flux (e), and surface sensible heat flux (f), for all days, cold days, and warm days of the 36 day simulation with 625 m grid spacing. In panels (a) and (b), solid lines show simulation data, and dashed lines show RADOLAN data. Averaging was done over all grid boxes where radar data are available.

For the 6 cool days, SCAI is in general larger, while COP is lower than the corresponding indices for the 36 days period (Fig. 9e,f), indicating the presence of more numerous and smaller objects. Although the degree of organization of these objects is weaker than for the full period (Fig. 9e–g), the variability in the shape (Fig. 9h) is similar to that in the larger period. In contrast to the cool days, during the 6 warm days, SCAI and COP show similar diurnal cycle to the 36 days period (Fig. 9i,j), revealing

5   the presence of fewer and larger objects, which favours organization. $I_{\mathrm{org}}$ also indicates a stronger degree of organization (Fig. 9k) in comparison with the cool days. Although $I_{\mathrm{shape}}$ is noisier on warm days, it also follows a similar behaviour (Fig. 9l) as seen during the longer period.

Overall, although the indices hint at an underrepresentation of convective organization and more compact objects in the 625 m LES, radar and model agree that organization is stronger on warmer days. However, there is a less clear signal for the warm

10   days compared to the average, than for the cool days.

## 4.3 Track statistics

We have shown in section 3.3 that in addition to the four convective organization indices, the rain cell tracking result provides information on the degree of organization in the three different model resolutions. In this section we apply the rain cell tracking

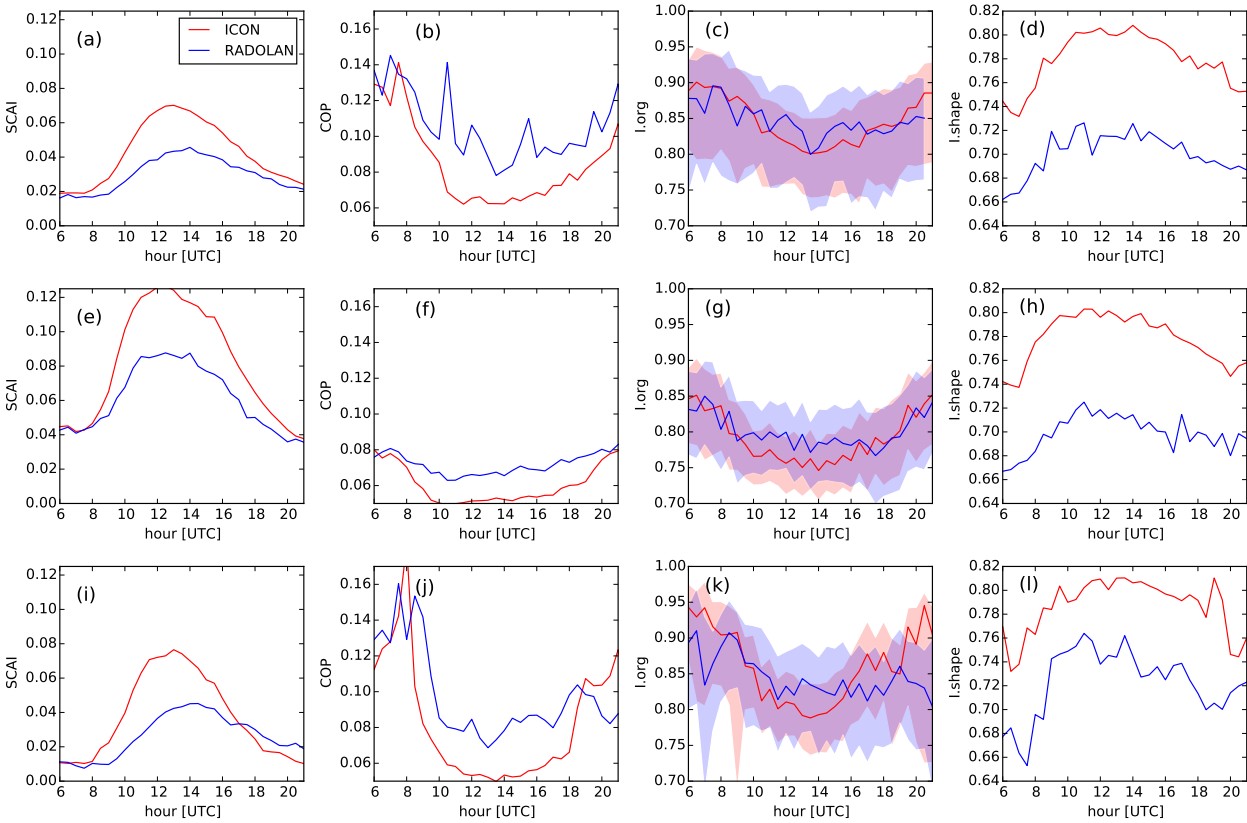

**Figure 9.** Mean diurnal cycles of the convective organization indices SCAI (a,e,i), COP (b,f,j), $I_{\mathrm{org}}$ (c,g,k; color shading shows the range between the 2.5th and 97.5th percentile as described in section 2.4), and $I_{\mathrm{shape}}$ (d,h,l), for all days (a–d), for cool days (e–h), and for warm days (i–l), for the model simulation with 625 m grid spacing, and for RADOLAN. Averaging was done over all grid boxes where radar data are available.

in a similar way on the 36-day continuous simulation with 625 m grid spacing with a separate analysis for the 6 cool days and 6 warm days. Table 2 shows that there is a consistent trend in the ratio of both of the number and the total precipitation produced by solitary tracks, and that this trend is the same for model and RADOLAN data: there is a smaller fraction of solitary tracks on the cold days and a larger one on the warm days, compared to the full simulation period. Likewise, the solitary tracks
5  contribute to a fraction of total rainfall that is smaller on cold days, but larger on warm days. This trend is weaker in the model than in the radar data. At first glance, this result seems to contradict our analysis of the three 3-domain days, where we argued that a *larger* contribution of solitary tracks corresponds to a *weaker* degree of organization: instead, the organization indices in Fig. 9 show *weaker* organization on the cold days, although the contribution of solitary tracks is *smaller* meaning that a larger fraction of tracks is subject to merging or splitting events. However, it should be kept in mind that there was also more total
10  precipitation in the analysis domain on the cool days, as compared to the total simulation period (Fig. 7), which is also reflected by the *total* number of tracks: while there are on average 21533 solitary tracks per day for the full model period, the number of

**Table 2.** Ratio of the number of tracks of given track types (solitary, tracks that involve only merging, tracks that involve only splitting, and tracks involving both merging and splitting), and the total amount of rainfall that they contribute, relative to the total number, and rainfall amount, respectively, of all tracks. Tracks that touch the domain boundaries are removed from the analysis. Fractions (in [%]), including all 36 model days, and conditioned on only the cold, and the warm days, as defined in Table A1, are given for both the model simulation (M), and the RADOLAN composite (R).

| Ratio (number; amount) [%] | All days (M) | Cool days (M) | Warm days (M) | All days (R) | Cool days (R) | Warm days (R) |
|---|---|---|---|---|---|---|
| Solitary tracks | 38.8; 12.1 | 35.5; 11.9 | 39.1; 13.6 | 29.8; 5.1 | 26.5; 4.1 | 35.7; 8.4 |
| Involving only merging | 23.1; 27.0 | 23.8; 27.4 | 24.1; 28.7 | 27.8; 25.1 | 28.9; 25.1 | 26.9; 28.5 |
| Involving only splitting | 27.1; 26.5 | 28.6; 28.3 | 27.1; 25.6 | 27.6; 24.2 | 28.7; 23.1 | 25.2; 27.8 |
| Both merging and splitting | 11.0; 34.4 | 12.1; 32.4 | 9.7; 32.1 | 14.8; 25.6 | 15.8; 47.7 | 12.1; 35.3 |

solitary tracks per day for the cold days was 33367 and therefore in total larger, while for the warm days there was a smaller number of only 20010 solitary tracks per day. For the RADOLAN data, these numbers were 8882 (all days), 17288 (cool days), and 9024 (warm days), respectively. Therefore, model and RADOLAN data agree on a larger *total number* of solitary tracks for the 6 cool days, in consistency with the hypothesis that a weaker organization on the cool days is associated with a larger

number of non-interacting rain cells. That the solitary track ratio with respect to the *total number* of all tracks is slightly smaller on the cool days, could be due to the fact that the larger number of precipitation objects (as indicated by the SCAI and COP indices) makes it more likely that neighboring objects interact with each other. This phenomenon was observed in the idealized LES study by Moseley et al. (2019) where model simulations with more convective rainfall and a larger number of rain cells showed a larger contribution of interacting rain cells to the total precipitation. We note that due to the large differences in the

total number of tracks between warm and cool days, the tracking statistics are more difficult to interpret here, compared to the more robust differences in track statistics between resolutions presented in Sec. 3.3.

    The box-and-whisker plots in Fig. 10 show the statistics of maximum track intensities, maximum cell radii, and total precipitation amount of the solitary tracks. The solid lines in Fig. 10a–c show that the above mentioned larger number of solitary tracks per day of the cold days (Fig. 10b) is distributed over all track durations. Compared to the total ensemble of all 36 days,

a smaller (larger) fraction of solitary tracks reach higher maximum intensities on cool (warm) days, and in consistency with the 99th percentile of rain intensities shown in Fig. 7b, there is a weaker temperature sensitivity seen for the model data as compared to RADOLAN. This intensification of the solitary tracks with temperature, especially for the tracks with a life time longer than 1 hour, can be seen even more clearly in the total amount of precipitation produced by the tracks (Fig. 10g–i). A dependence of the cell sizes reached by solitary tracks in temperature is less clear (Fig. 10d–f).

To briefly summarize the tracking result in this paragraph, we find that solitary tracks of comparable duration can reach higher precipitation amounts on warm days as compared to cool days. This shows an intensification of solitary convective rain tracks with temperature. However, this intensification is found to be weaker for the model data compared to RADOLAN. Furthermore, a larger number of solitary tracks on cool days in both model and RADOLAN data is consistent with a weaker degree of convective organization.

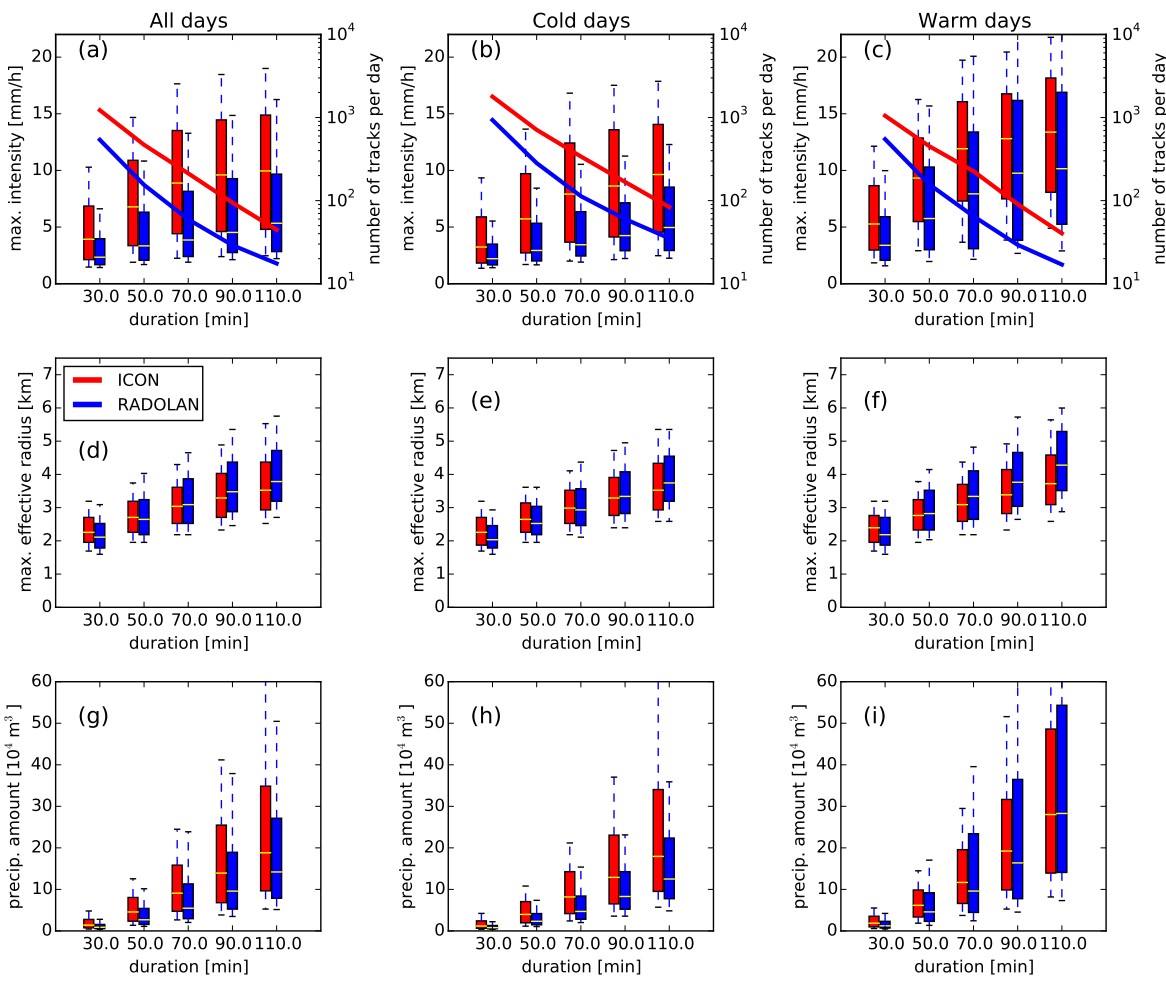

**Figure 10.** Box-and-whisker plots showing the statistics of solitary tracks for the whole 36-days period, for all (a,d,g), cold (b,e,h), and warm (c,f,l) days. Values of track maximum intensity (a–c), and track maximum effective radius (d–f), and total amount of precipitation produced by the individual tracks (g–l), are conditioned on track duration ranging between 20 and 120 min, in 5 bins of 20 min width. Boxes indicate the 25th and 75th percentiles and the median (red bar), whiskers indicate the 10th and 90th percentiles. The number of tracks in each bin is indicated by the solid lines in panels a–d (note the logarithmic axis on the right).

## 5 Discussion

We have evaluated the impact of horizontal resolution on explicitly simulated convective precipitation, and analysed the sensitivity of convective organization to daily mean 2m air temperature on the 36-day continuous simulation with 625 m grid spacing. The impact of horizontal resolution is significant. Our study indicates that compared to the RADOLAN data, the diurnal cycles, life cycles, and degree of convective organization is simulated better at the innermost nest with 156 m horizontal grid spacing. This is in agreement with previous studies which argued that for a sufficient resolution of the processes within deep convective updrafts, models with grid spacing of the order of ca. 100 m are required (Petch et al., 2002; Bryan et al., 2003), and that there is neither bulk convergence nor structural convergence at coarser resolutions (Panosetti et al., 2019). At 625 m and to a smaller degree at 312 m grid spacing, convection tends to set in too rapidly, and many isolated deep convective cells are scattered over the domain. In contrast, at 156 m, we find a smoother onset of convective updrafts with lower peak intensities, and a stronger degree of organization, that in general shows a better match with the radar data. In addition, the tracking analysis revealed that the stronger organization of the higher resolved simulations is accompanied by an increased tendency of convection to form larger clusters: the 156 m simulation shows a lower number of isolated rain cells, and their contribution to total rainfall is lower. Vice versa, the total contribution of the tracks that undergo merging and splitting is clearly higher for the higher resolved simulations. Petch et al. (2002) argues that at coarser resolutions the models fail to compensate for the lack of resolved transport out of the sub-cloud layer, leading to a delayed spin-up of convection relative to that obtained in the better-resolved simulations. This delay in the spin-up might then lead to the too explosive convective initiation that we find in our analysis. We speculate that this could also be the reason for the suppressed organization of the 625 m simulation compared to radar: as soon as a convection cell is initiated, it is already fully developed and therefore does not have enough time to interact with neighbouring cells within its life time. However, this is a hypothesis that should be tested in a future study. Such a study should investigate the processes that happen within merging cells in more detail.

An improved subgrid scheme might lead to more realistic results and a decreased sensitivity to resolution, while the Smagorinsky subgrid scheme used in our model seems to be not the optimal choice at 625 m grid spacing, as some of the larger boundary-layer eddies are likely unresolved. We also note that the microphysics scheme might have significant impacts on organized convection. An analysis of the impact of different physical parametrizations on the simulated convection is not covered here, and we encourage future studies in this direction.

Similar to Pscheidt et al. (2019), we find in general a too large number of small clouds as indicated by the rain cell size distribution, and also by the SCAI and COP indices in the model simulations, but in contrast to their findings we see a tendency towards fewer and larger objects at high resolution which we find more realistic as evaluated against the RADOLAN data. Further, similar to our study, Pscheidt et al. (2019) find that objects are more elliptic at higher resolution as indicated by the $I_{\mathrm{shape}}$ index. However, they find this to be less realistic as compared to RADOLAN and satellite data, while in two of the three days that we analyzed, $I_{\mathrm{shape}}$ at the 156 m nest matches better with RADOLAN. Although Pscheidt et al. (2019) use the same model at the same resolutions, and partly the same observational data as in our study, they have analyzed different model days. Thus, the reason for the discrepancies might be that differences among different model resolutions depend on synoptic

situations, which indicates that a larger sample of model days is needed to confirm the hypothesis that convective organization is better simulated at 156 m grid spacing. However, our hypothesis is also supported by the tracking result which shows that there are less solitary tracks (which – in turn – means more interaction between tracks) at higher resolutions, which provides a better match to the RADOLAN data. Pscheidt et al. (2019) recommend that COP and SCAI can be replaced by object sizes and object number, respectively, since they are mainly influenced by these two quantities. However, supplementary information on the degree of organization is provided by $I_{\text{shape}}$ and $I_{\text{org}}$, in particular since the latter is able to distinguish between three possible categories: organized, regular, and random. Our study confirms this hypothesis, with the addition that tracking objects in time can give valuable information on the tendency of convection to form clusters. Another possible improvement could be new indices that take into account both ends of the size distributions function separately: Neggers et al. (2019) have shown that spatial organization affects both ends of the cloud size PDF, but in different ways: while the number of large clouds increases, there is an enhanced variability in the number of small clouds, especially shallow cumulus clouds below the 1 km scale. However, in our study, we are mainly concerned with deep precipitating convection where such small cloud sizes are neglected.

Consistent with theory, our analysis of the continuous 36-day period with 625 m grid spacing shows that convection gets more intense with higher near-surface temperatures. A separate analysis of 6 cool (below 16°C) and 6 warm days (19–21°C) shows a consistent increase with temperature in the 99th percentile of precipitation intensity, as well as in the total amount of precipitation generated by solitary tracks. This finding is encouraging, since it confirms that the increase of extreme precipitation with temperature can be represented with CRMs at the kilometer scale. However, in our simulation period the simulated increase from cool to warm days is smaller in magnitude than in the RADOLAN data. In addition to heavy precipitation intensities, we also find a temperature sensitivity of the convective precipitation indices: in particular, they show a weaker degree of organization for the cool days in both model and RADOLAN data. Although this is consistent with a larger number of solitary tracks on the cool days, the *fraction* of solitary tracks is smaller on the cool days. This is probably due to the fact that although the degree of organization might be weaker, there was more total precipitation on the cool days in our simulation period, making an interaction of precipitation objects more likely since they are on average closer together. The large differences in the total number of tracks between warm, cool, and all days in the analysis of the temperature sensitivity makes the interpretation of the tracking result more difficult, compared to the resolution analysis. A deeper investigation of the interaction between events is left to a future study, and the idealized study by Moseley et al. (2016) suggests that interaction between cells might well be intensified with higher temperatures. Our study also cannot answer the question open if higher resolution will lead to an improved simulation of the sensitivity of heavy rainfall and convective organization to temperature, as too few model days on all 3 nests are available. Given that the magnitude of the intensification of heavy rainfall with temperature has both a thermodynamic (based on the CC argument) and a dynamic aspect, and that thermodynamic processes can be expected to be rather independent of resolution, we can assume that it is mainly an insufficient representation of the dynamics within the convection cells that causes an underestimated intensification at 625 m grid spacing. Although, in contrast to our results, Ban et al. (2014) do not report an underestimated temperature sensitivity of heavy rainfall in Switzerland with a 2.2 km model, the

strong orography in their study region is absent in the largest part of our simulation domain, such that a direct comparison to our study may be difficult.

In addition to these findings, we have shown that the Iterative Raincell Tracking method (IRT) is not only a useful tool to study the life cycles of isolated convective rain events (that is, solitary tracks), but it is also able to provide information on the convective organization in the model simulations and observational data. In general, a smaller total contribution of isolated cells to the total rainfall indicates that the tendency of convection to interact and form clusters is larger, since it means that a larger fraction of tracks experiences merging and splitting. Therefore, our tracking result is consistent with the convective organization indices. However, as also stated by Rasp et al. (2018), these indices describe only the spatial structure of the convection, but neglect the temporal structures of convective memory, which is an import aspect for parametrizations. Therefore, there is the need for new types of indices that also involve information on the temporal evolution of convective organization. A further development of our tracking method may fill this gap, as it includes the time evolution of convection cells and therefore has the potential to provide a more comprehensive description of the processes that happen when repeated merging of individual convection cells lead to large clusters, such as mesoscale convective systems, squall lines, and tropical cyclones.

Finally, we mention another important aspect that we have not addressed in the present study, but which however has been shown to have an important impact on convective organization, namely cold pools: when cold pool gust fronts collide, they sometimes trigger another convective precipitation cell, leading to a complex feedback between the convective rain cells and the cold pools that they generate (Haerter et al., 2019). Cold pools are clearly visible in our model data, and an analysis of their role in triggering new convection cells will be published in a separate paper (Hirt et al., 2020).

## 6 Conclusions

Based on a 36 day long continuous simulation for May and June 2016, we have shown that ICON in a limited area setup over Germany with a grid spacing of 625 m is able to simulate an intensification of isolated convective rain cells with temperature. However, the magnitude of the simulated intensification is smaller than shown by the RADOLAN radar composite. Further, we find a weaker degree of organization especially on cooler days, which is reflected by the convective organization indices, but also by a larger number of non-interaction (solitary) rain cell tracks.

An analysis of the three days that are available on all three nests showed that the convective organization pattern is best simulated at the highest resolution with 156 m grid spacing. At the coarsest nest with 625 m grid spacing, we find that convective events are too strong at the beginning of their life cycles, that they are weaker organized, and that they show a weaker tendency to merge and form clusters. This indicates that not all processes in the convective updrafts are optimally resolved at this resolution. Overall, our evaluation of the three model resolutions suggests that an increase of model resolution toward the 100 m scale has the potential to provide a more realistic simulation of convection.

Based on our finding that stronger convective organization is associated with a smaller number of non-interacting tracks and more merging and splitting events between objects, we propose the development of new convective organization indices

**Table A1.** Mean 2 m temperature $T$ and 10 m wind speed $v$ for each day, averaged over COSMO analysis data between 8 and 20 UTC, and daily precipitation sums $P_M$ from model output, and $P_R$ from RADOLAN. In the temperature column, colors show days classified as cool (blue), warm (orange), and very warm (red). The days marked in yellow are simulated on three domains. The table on the right continues the left.

| Mon–Day | $T$ [$^\circ C$] | $P_M$ [$mm$] | $P_R$ [$mm$] |
|---------|------------------|--------------|--------------|
| 05–26 | 16.9 | 1.58 | 0.80 |
| 05–27 | 17.7 | 3.50 | 3.88 |
| 05–28 | 18.9 | 3.46 | 4.71 |
| 05–29 | 18.2 | 8.32 | 11.11 |
| 05–30 | 17.4 | 9.11 | 10.98 |
| 05–31 | 17.5 | 3.93 | 5.06 |
| 06–01 | 17.2 | 10.48 | 9.35 |
| 06–02 | 17.1 | 5.09 | 7.43 |
| 06–03 | 18.1 | 4.7 | 4.63 |
| 06–04 | 19.5 | 3.07 | 3.59 |
| 06–05 | 20.0 | 3.28 | 4.55 |
| 06–06 | 20.5 | 1.72 | 1.49 |
| 06–07 | 20.7 | 2.28 | 3.47 |
| 06–08 | 17.8 | 3.67 | 5.21 |
| 06–09 | 16.5 | 1.72 | 1.56 |
| 06–10 | 17.3 | 0.14 | 0.12 |
| 06–11 | 16.4 | 2.96 | 3.68 |
| 06–12 | 15.9 | 7.01 | 8.96 |

| Mon–Day | $T$ [$^\circ C$] | $P_M$ [$mm$] | $P_R$ [$mm$] |
|---------|------------------|--------------|--------------|
| 06–13 | 15.7 | 7.36 | 8.00 |
| 06–14 | 15.9 | 6.07 | 6.71 |
| 06–15 | 15.8 | 6.19 | 7.04 |
| 06–16 | 17.0 | 5.19 | 7.42 |
| 06–17 | 15.4 | 11.12 | 12.03 |
| 06–18 | 16.2 | 3.31 | 3.68 |
| 06–19 | 15.3 | 2.31 | 1.71 |
| 06–20 | 17.2 | 7.50 | 6.63 |
| 06–21 | 18.4 | 1.20 | 0.70 |
| 06–22 | 22.0 | 0.30 | 0.11 |
| 06–23 | 26.0 | 2.77 | 4.07 |
| 06–24 | 24.1 | 3.12 | 6.93 |
| 06–25 | 19.6 | 11.11 | 15.09 |
| 06–26 | 17.2 | 2.11 | 2.02 |
| 06–27 | 17.3 | 2.35 | 1.67 |
| 06–28 | 18.5 | 1.50 | 1.33 |
| 06–29 | 19.3 | 2.82 | 4.14 |
| 06–30 | 18.6 | 2.89 | 3.55 |

that are capable of monitoring not only the spatial, but also the temporal evolution of the convective clustering process. Such indices could be based on existing tracking algorithms such as the IRT method that we applied within this study.

*Code and data availability.* Primary data and scripts used in the analysis that may be used for reproducing the authors' work will be stored in the DKRZ long term archive. (A permanent URL link can be provided after the manuscript is accepted.)

# Appendix A: Overview of simulated days

*Author contributions.* R.H., G.C., and C.M. designed and performed the model simulations, C.M. performed the tracking analysis, I.P. calculated the convective organization indices. C.M. conceived the original idea and coordinated the analysis. All authors contributed to the manuscript.

5  *Competing interests.* There are no competing interests.

*Acknowledgements.* We thank the German Climate Computing Center (DKRZ) for providing computing resources and assistance with the performance of the ICON simulations. The Germany Weather Service (DWD) is acknowledged for providing the RADOLAN radar composite. We thank Sophia Schäfer, Cathy Hohenegger, and two anonymous referees, for proof reading the manuscript and suggestions for improvement. We acknowledge funding through the German Federal Ministry of Education and Research for the project
10  "HD(CP)$^2$ - High definition clouds and precipitation for advancing climate prediction" within the framework programme "Research for Sustainable Development (FONA)", under 01LK1506F, 01LK1501B, 01LK1507B, and 01LK1507A.

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
