# Peer review of "Impact of resolution on Large Eddy Simulation of mid-latitude summer time convection"

_Atmospheric Chemistry and Physics, 2019_

## Referee Comment (RC1) · Anonymous Referee #1 · 20 Sep 2019

This articles discusses the impact of resolution on the organisation of convection in a LES of summer time convection over Germany, as well as the sensitivity of precipitation to 2m temperature in simulations with 625m grid spacing.

It concludes that there is a benefit in using a simulation with 156m grid spacing as compared to 625m in terms of the diurnal cycle of convection and some of the measures of convective organisation, and that the model underestimates the sensitivity of rainfall to 2m temperature.

Most of the analysis is a valuable analysis of ICON-LEMs representation of summertime convection. I have some questions about both the methodology and the conclu-

sions, and some revisions will be require to make the manuscripts suitable for publication.

The writing is mostly clear, although some of the sentences are rather long and the language could be more concise at points (I have suggested some changes here, but more could be made). There are also places where e.g. including hyphens would make the text more readable.

General comments:

- Title: "resolution and air temperature" -> I find this a bit confusing, as resolution is determined by the model configuration, but air temperature is not a model parameter (it impacts the simulated convection, rather than the simulation itself). Maybe mention "sensitivity to 2m temperature" specifically?

- It would be good to add some further information about earlier studies that have looked at sensitivity of convection to resolution. One term that has come up in recent years is so-called bulk-convergence (i.e. the convergence of larger-scale mean properties) as opposed to structural convergence (e.g. Langhans et al 2012, https://journals.ametsoc.org/doi/full/10.1175/JAS-D-11-0252.1, Panosetti et al 2019, https://rmets.onlinelibrary.wiley.com/doi/full/10.1002/qj.3502).

- p4, l26: The authors mention they have resampled their results on a larger grid. Although such a resampling is a good idea, it is important to be aware that the method used may influence the results. For example, it is likely that the cloud fraction increases due to the resampling, because some grid cells will only partially meet the threshold (this is certainly the case if non-zero liquid water would be used as the mask). It is not fully clear to me how this can be prevented, but it may be worth describing the possible effects. One alternative strategy for regridding would be to randomly sample one of the columns: this would keep the cloud fraction the same (statistically). This may also be relevant to the track statistics (section 3.3.)

- Section 2.4: It would be good to add some more information on the interpretation of and differences between some of the indices of convection, such as SCAI and COP. It is not clear to me what the advantages of using one metric over the other would be from the current description.

- One of the metrics which is currently missing, and may be helpful in terms of the interpretation of the other indices, is a probability distribution function of object sizes in each simulation and the radar. This could potentially be plotted both for the original data and the resampled data.

- One potential issue with some of the metrics, e.g. Iorg, may be that it can give disproportionally high importance to smaller objects. One option here would be to consider a measure of organisation that considers objects of the same size (see e.g. Neggers et al 2019, https://journals.ametsoc.org/doi/full/10.1175/JAS-D-18-0194.1). It would be good to mention this in the text.

- In section 3.4, there seems to be a significant difference between all simulations and the radar in terms of the organization indices. It would be good to investigate the cause of this in more detail, for example by looking at object size distributions, or the original fields from which the indices were derived.

- One striking feature of figure 3 is that the development of SCAI looks different between different days. The other metrics seem to have a very similar development on different days, and for COP and I-org, the differences between radar and simulations are of the same order as the differences between the development of the indices on different days. This may point to the SCAI being more useful than some of the other indices.

- One aspect of SCAI that I am wondering about is the fact that it seems to be consistently low at night. This may partially be due to organised propagating systems, but I am also wondering how the SCAI behaves when convection is (almost) absent? Is there a strong correlation between SCAI and cloud cover?

- What explains differences in night-time behaviour between cool and warm days?

- The results should likely be interpreted in the context of a given configuration. It would be worth stressing that changes to e.g. the microphysics scheme, as well as further changes to the turbulence scheme mentioned already, will impact on the results. - In figure 8, again there seem to be differences between radar and ICON in terms of I-shape and COP, which are bigger than the differences between warm and cool days. Do you understand what causes these differences?

- p19, l11: "larger clusters". I am not sure if this can be said on the basis of the statistics provided. Can you clarify?
* * *
Minor/editorial issues (these are mostly easy to address, but could improve the presentation quality):

- p1, l4-5: "showing a considerable..most of the days" -> this is a long clause, maybe it can be broken up?

- p1, l8: "showed that"

- p1, l14: diurnal cycles -> "the diurnal cycle" is clearer, I think.

- p1, l16: "CRMs" (plural)? Or refer to the technique instead.

- p1, l16: "necessary" -> I would simply say "suitable", possibly a well-designed parametrisation could also have the correct diurnal cycle.

- p1, l21: it would be good to explain the differences between LES and kilometre-scale modelling in terms of the turbulence scheme.

- p2, l1: this sentence is on the long side.

- p2, l10: the cumulus scheme would be worth mentioning here as well.

- p2, l16: the presence of super-CC scaling may depend on the method of analysis (Ban et al. 2015, https://agupubs.onlinelibrary.wiley.com/doi/full/10.1002/2014GL062588).

- p2, l16-17: "even" occurs twice here.

- p2, l20: there are earlier references on the dynamic and thermodynamic components of this sensitivity.

- p2, l22: "air temperature" is a somewhat vague term. I would probably mention 2m temperature specifically, unless a different level/set of levels is used.

- p2, l24: "above-mentioned"

- p2, l27: "object-oriented"?

- p3, l5: "by" -> "of"?, "suited to provide" -> "provides"

- p3, l9: remove "applied"

- p3, l10: 165 -> 156?

- p3, l15: remove "implemented".

- p3, l22: place a comma after "work".

- p3, l26: it may be good to mention something about the surface layer parametrization and the (absence of?) a subgrid-scale cloud scheme. For the turbulence scheme, it would be good to mention how grid anisotropy is dealt with.

- p3, l29: "down scaled" -> I am not sure if this is the right verb.

- Section 2.2: it would be good to spend some text on model initialisation and spin-up of convective structures for the inner nests.

- p4 l4: article "the" missing before billions

- p4, l5: do you mean "the first days"?

- p4, l8: comma after "costs"

- p4, l11: the differences are also partially due to the inherent unpredictability of convection

- p4, l14: "wide spread" -> "widespread"

- p4, l14: can you give some more information on the presence of cold pools during these days.

- p4, l23: "large scale" -> "large-scale"

- p5: l4 "time-interpolated"?

- p6, l25: note that I-shape is sensitive to discretisation: for example, as far as I can tell, a circle that is approximated by a large number of squares would have a shape ratio of pi/4, rather than 1.

- p6, l27: "contour"

- p7, l4: "defined...results" -> this a very long clause, it would be better to split it.

- p7, l31: "different resolutions" (plural)

- p7, l32: "the" (capitalisation)

- p8, l1: are you referring to June 6, instead of June 3, here?

- p8, l1-5: the later onset in the simulations with higher resolution appears to be consistent on June 3 and June 6. However, some of the other differences may be due to individual large storms. It would be good to mention this at least (running ensemble forecasts of the lowest resolution run would help to establish this internal variability, though I am aware this may be a major effort).

- In section 3, the terms organisation and clustering are used somewhat interchangeably (it may be good to make explicit which of the measures identify clustering in particular, i thought this was mainly I-org).

- p8, l16: "somehow different" -> remove "somehow", explain the differences for June 6.

- Caption fig. 3: 165 -> 156

- p9, l10: "We now how...tracks" -> I would rephrase this sentence, to tell more about the kind of additional information provided, rather than the fact that additional information is provided.

- p9, l10: "are provided" -> "is provided"

- p10, l2: there is an issue with the parentheses here.

- p10, l5 "even" -> "event"

- p10, l15: remove italics here (m) for consistency (same applies to e.g. p11, l16 and p13, l1/12/18)

- p10, l17: could the relative percentages be affected by the regridding method?

- p10, l20: "composites"

- p10, l24: "sizes" (capitalisation)#

- p11, l1: "at" -> "in"

- p11, l4 "(g-i)"

- p11, l9: it would be good to add a subscript to the areas, and put "A" in italics.

- p11, l20: "including" -> "and for"

- p11, l30: it would be good to refer back to the concept of bulk-convergence here.

- p12, l8: in terms of differences between precipitation between model and forecast, some of these may be due to the uncertainty in boundary conditions.

- p13, l4 "added value" (no article)

- p13, l10: "6-day period"

- p13, l14: "introduction" (capitalisation)

- p14, l9: "than for the mean"

- p14, l12: "introduction" (capitalisation)

- p14, l14: is an underestimation of sensitivity to temperature sensitivity consistent with previous findings? It is not clear to me if this result is significant, given that only a few rain cells may have a big impact on the 99th percentile rainfall. The sensitivity test here is not a very strong one, as the bulk of the underlying data stays the same. One option would be to look at how much this differs between (subsets of) individual days in each category (looking at the diurnal average 99th percent rainfall).

- p15, l10: "similar to the larger period" -> "similar to that in the larger period"

- p16, l1: "a stronger degree"

- p17, l2: "there" (capitalisation)

- p17, l2: the trend is consistent, but the actual number of solitary tracks is very different. The differences between "cool days" and "all days" also seem more pronounced in the radar data.

- p17, l3/l5/l9: "there","instead","while" (capitalisation)

- p17, l4-19: may this impact on the interpretation of the 3-domain days as well (in the light of the remark about regridding in the general introduction)?

- p17, l20: "Similar as" -> replace by a construction with "similar to"

- p17, l24: "temperaure" -> "temperature"

- p18, l2: "the longer durations tracks above 1 hour life time" -> "the tracks with a life time longer than 1 hour"

- Figure 5 and 9 have intensity with units km. This should likely be mm/hr.

- p18, l4: "on warm days" is repeated here.

- p19, l4: see my comments on the title. These are different kinds of sensitivities (to the model configurations, to the atmospheric state).

- p19, l15: it could be good to cite some work on turbulence schemes for the boundary-layer grey-zone here.

- p19, l18: "should be left" -> "will be left"

- p19, l19: "similar as" -> "similar to"

- p19, l20: "fewer and larger objects"

- p19, l23: "as to compared to" -> "as compared to"

- p19, l31: "at least simulated qualitatively" -> it would be good to rephrase this (the word choice/order here is odd: you could say the sensitivity to temperature has the same sign)

- p20, l3: "we","in" (capitalisation)

- p20, l7: "objects" (plural)

- p20, l8: "may be left" -> "is left"

- p20, l11: "higher-resolved" -> or change formulation?

- p20, l28: "in May and June 2016" -> "for May and June 2016"

- p20, l30: but to a smaller degree. -> the sentence construction needs to be changed here (see also my previous comment on significance).

- p21, l1: Remove "However"

- p21, l5: place a comma after "Overall".

---

## Referee Comment (RC2) · Anonymous Referee #2 · 27 Sep 2019

This paper examines the convection activity simulated by the ICON-LEM at 625 m horizontal resolution for 36 days in summer over Germany and compares it to that observed by the ground radar system. It also examines the impact of horizontal resolution on the simulated convection over 3 days by nesting into 300 and 100 m resolutions. I agree with the other reviewer that the authors showed great expertise in terms of deployment of the model and the analyzing techniques. But the logic of the paper, in other words, why they set up and present the study in such a way to answer the questions they want to answer, is not clear enough to me. I suggest a major revision. Here are my major concerns,

1. In the analysis of the 36-day simulations, the authors separated the cases into warm and cool days to compare the impact of surface air temperature on convection activity as seen in simulations and in observation.One of the motivations for doing this seems to be that the authors are concerned with the ability of the cloud-resolving models to correctly simulate the response of atmospheric convection in a warming climate, which they hinted at in the introduction. But it is not obvious to me that the contrast in the large-scale environment between the warm and cool days chosen in this study is comparable to the contrast in typical large-scale environment of middle-latitude convection activity between the current and future warmer climate. If the authors think they are comparable, they should make the claim more explicitly. If the authors just want to compare the sensitivity of the simulated convection to different environmental conditions as characterized by the surface air temperature the analysis is completely valid in my opinion given the importance of surface condition for summertime convection over mid-latitude land.

2. I wonder what criterion the authors used for selecting cases to perform higher-resolution simulations and why the authors did not choose those cases so that they could also investigate the contrast between warm and cool days at higher resolutions (even just with 2 or 4 cases).

---

## Author Comment (AC1) · 9 Oct 2019

Thank you for carefully reading our manuscript, and your suggestions for improvement. Before the open discussion phase ends, we here provide preliminary replies to your major comments. For easier readability, and for referencing, we have attached numbers to your general comments.

General comments:

––––––––––––––––––

1. Title: "resolution and air temperature" -> I find this a bit confusing, as resolution is

determined by the model configuration, but air temperature is not a model parameter (it impacts the simulated convection, rather than the simulation itself). Maybe mention "sensitivity to 2m temperature" specifically?

Reply: We suggest to change the title to: "Impact of resolution on Large Eddy Simulation of mid-latitude summer time convection", thus leaving out the temperature sensitivity.

————————-

2. It would be good to add some further information about earlier studies that have looked at sensitivity of convection to resolution. One term that has come up in recent years is so-called bulk-convergence (i.e. the convergence of larger-scale mean properties) as opposed to structural convergence (e.g. Langhans et al 2012, https://journals.ametsoc.org/doi/full/10.1175/JAS-D-11-0252.1, Panosetti et al 2019, https://rmets.onlinelibrary.wiley.com/doi/full/10.1002/qj.3502).

Reply: Thank you for drawing our attention to these studies. Panosetti et al. (2019) write that they find neither bulk-convergence nor structural convergence over Germany at the 1 km grid spacing scale. Their highest resolution simulation has 550 m grid spacing which is close to our outer ICON-LEM nest with 625 m grid spacing. Although we analyze all simulations on the same grid, our study mainly addresses structural convergence, as it is mainly concerned with the shape, time evolution, and organization of individual convection cells, rather than mean quantities averaged over areas that are larger than individual clouds. Our conclusion is that there is no structural convergence at grid spacings that are coarser than the 100 m scale, which is consistent with Panosetti et al.. In the revised manuscript, we will mention the concept of bulk- and structural convergence in the Introduction, and refer to it in the discussion of our results.

————————-

3. p4, l26: The authors mention they have resampled their results on a larger grid. Although such a resampling is a good idea, it is important to be aware that the method used may influence the results. For example, it is likely that the cloud fraction increases due to the resampling, because some grid cells will only partially meet the threshold (this is certainly the case if non-zero liquid water would be used as the mask). It is not fully clear to me how this can be prevented, but it may be worth describing the possible effects. One alternative strategy for regridding would be to randomly sample one of the columns: this would keep the cloud fraction the same (statistically). This may also be relevant to the track statistics (section 3.3.)

Reply: There are several reasons why we applied a regridding. The primary reason is that the original ICON-LEM output is given on an unstructured triangular grid. The codes for the calculation of the indices, and the rain cell tracking, need a regular lat-lon grid as input, and an extension of our codes to handle the unstructured raw data would be a difficult task. The second reason is that we prefer to compare the data of the three different model resolutions, and the radar data, on the same grid, to reach a fair comparison. We chose a 1x1 km lat-lon grid, since this is roughly the resolution of the radar data. Further, it is only slightly coarser than the resolution of the coarse ICON-LEM resolution with 600 m grid spacing of the triangle edges. However, as the effective resolution of the ICON-LEM data is larger than the grid spacing, we can assume that there is no loss in resolution at least for the 600 m simulation. A similar regridding has also been used for other studies which also analysed ICON-LEM output, like Heinze et al. (2017), and Pscheidt et al. (2019).

As you point out, there could be some problems that the cloud edges are not clearly defined, since the resampling may lead to lower values such that the threshold may not be reached at some grid boxes. However, since we track the surface precipitation field, with a relatively low threshold of 1 mm/h rain intensity, we assume that this effect is rather small at a resolution of 1 km. In Moseley et al. (2013) (https://doi.org/10.1002/2013JD020868) it was shown that at least for radar data over

Germany, the rain cell tracking result is not largely different if the resolution is 1 km or 2 km. We will mention this is the revised manuscript. The conservative remapping method also takes care that the total amount of rainfall is not changed, which is not the case with a random sampling of columns.

________________-

4. Section 2.4: It would be good to add some more information on the interpretation of and differences between some of the indices of convection, such as SCAI and COP. It is not clear to me what the advantages of using one metric over the other would be from the current description.

Reply: A more detailed discussion of the organization indices and the differences between them is given in Pscheidt et al. (2019). This study uses the same simulation domain as our paper, and partly the same data. Therefore we can refer to it for more information on the indices, their general behavior over the study domain, and how they should be interpreted. The purpose of our present study is mainly to show which of these indices are affected mostly by model resolution, and investigate their sensitive to daily mean temperature. Another main purpose of our study is to show that the rain cell tracking method can add additional information on the temporal evolution of convective organization that the indices can not provide, since they only "see" the spatial distribution of convection at instantaneous time records. For these reasons we decided to calculate and present all four indices SCAI, COP, I.org, and I.shape in the paper.

In their conclusions Pscheidt et al. write that since COP and SCAI are mainly influenced by the areas and the number of objects, respectively, they recommend to use I.org, I.shape, object area, and object number for characterizing the state of the spatial organization of the convection field. They also claim that I.org is in some respects superior to COP and SCAI, since it is able to distinguish between three possible categories: Organized, regular, and random. We will include this in the Discussions section of our paper.

[Figure]

———————————-

5. One of the metrics which is currently missing, and may be helpful in terms of the interpretation of the other indices, is a probability distribution function of object sizes in each simulation and the radar. This could potentially be plotted both for the original data and the resampled data.

Reply: We will plot some PDFs of cloud sizes for the three different model resolutions, and for the radar data, respectively. We also plan to include an additional plot comparing the size distributions in the revised manuscript. As we did not save the original data but only the regridded data-sets, we can perform this analysis only on the regridded data.

———————————-

6. One potential issue with some of the metrics, e.g. Iorg, may be that it can give disproportionally high importance to smaller objects. One option here would be to consider a measure of organisation that considers objects of the same size (see e.g. Neggers et al 2019, https://journals.ametsoc.org/doi/full/10.1175/JAS-D-18-0194.1). It would be good to mention this in the text.

Reply: Thank you for recommending this interesting study. Neggers at al. write that spatial organization affects both ends of the cloud size PDF, but in different ways: While the number of large clouds increases, there is an enhanced variability in the number of small clouds. This suggests that it is probably possible to extract more information on the spatial organization of convection, if new indices can be defined that take into account smaller and larger clouds separately. However, in our study, we are mainly concerned with deep precipitation convection where cloud sizes below 250 m are neglected, while Neggers at al. look also at smaller shallow cumulus clouds.

We will discuss this possible shortcoming of the indices in the Discussion section in the revised manuscript with reference to the study by Neggers at al.. However, the

improvement of existing indices for the spatial distribution of convection is not the main purpose of our study (see reply to your comment 4), and a subsampling analysis similar to Neggers et al. would be beyond the scope of the paper.

————————-

7. In section 3.4, there seems to be a significant difference between all simulations and the radar in terms of the organization indices. It would be good to investigate the cause of this in more detail, for example by looking at object size distributions, or the original fields from which the indices were derived.

Reply: As stated in our reply to your comment 5, we are going to plot some size distributions for the 3 resolutions and the radar data, and discuss the differences, and their possible impact on the indices. Are you referring to Fig. 3 (as there is no section 3.4 in the manuscript)?

————————-

8. One striking feature of figure 3 is that the development of SCAI looks different between different days. The other metrics seem to have a very similar development on different days, and for COP and I-org, the differences between radar and simulations are of the same order as the differences between the development of the indices on different days. This may point to the SCAI being more useful than some of the other indices.

Reply: As we mentioned in our reply to your comment 4 with reference to Pscheidt et al. (2019), SCAI is mainly influenced by the number of objects. The strong differences in SCAI between different days thus hint to different numbers of convection cells on the different days. On the other hand, the observation that COP and I.org are more robust among the days could mean that the size distributions do not vary that much among days.

————————-
* * *
Interactive
comment

9. One aspect of SCAI that I am wondering about is the fact that it seems to be consistently low at night. This may partially be due to organized propagating systems, but I am also wondering how the SCAI behaves when convection is (almost) absent? Is there a strong correlation between SCAI and cloud cover?

Reply: As SCAI is strongly affected by the number of objects, and the number of objects is larger when there is strong convection, it seems plausible that SCAI is larger at noon time when convection sets it. At night, precipitation is mainly large-scale, with larger, but fewer objects and weaker intensities. SCAI already begins to decline in the afternoon hours, which reflects the observation that convection begins to organize and forms larger objects due to merging, thus reducing the number of objects.

We did not store the cloud cover field fields, but we can check if there is any correlation of SCAI with the cloud liquid water field. However, we point out that our study focusses on daytime convection, since it is known that the nocturnal boundary layer is not sufficiently resolved at LES resolutions of 100 m and coarser, which may introduce unknown biases in cloud cover at night. See e.g. van Stratum and Stevens (2015) https://doi.org/10.1002/2014MS000370. We will mention this paper in the revised manuscript.

———————————-

10. What explains differences in night-time behaviour between cool and warm days?

Reply: As stated in our previous reply, we cannot make any statements on night time precipitation from our LES simulation, since the nocturnal boundary layer is unresolved. Therefore we condition or analysis on daytime temperatures between 8 and 20 UTC, only. To make this clearer, we will limit our plots to the time range between 6 and 21 UTC in the revised manuscript.

———————————-

11. The results should likely be interpreted in the context of a given configuration.

It would be worth stressing that changes to e.g. the microphysics scheme, as well as further changes to the turbulence scheme mentioned already, will impact on the results. - In figure 8, again there seem to be differences between radar and ICON in terms of Ishape and COP, which are bigger than the differences between warm and cool days. Do you understand what causes these differences?

Reply: As we neither have sensitivity studies with different turbulence schemes, nor with changed microphysical parametrizations, we cannot make any statements on the impact of different schemes on our results. We agree that there might be significant impacts on organized convection. We will make this more clear in the discussion section, and encourage further studies in this direction.

The lower COP and higher I.shape in the model data as compared to radar hint at an under representation of convective organization and more compact objects in the 625 m LES, respectively. However, radar and model agree that organization is stronger on warmer days. The reason for the suppressed organization could be related to the too explosive convective initiation at coarser resolutions as discussed on p. 19 l. 9-18. As soon as a convection cell sets in, it is already fully developed and does not have enough time to interact with neighbouring cells within its life time. However, this is a hypothesis that should be tested in a future study. Such a study should investigate the processes that happen within merging cells more deeply.

————————————-

12. p19, l11: "larger clusters". I am not sure if this can be said on the basis of the statistics provided. Can you clarify?

Reply: This statement mainly refers to the interpretation of the tracking analysis, notably the numbers shown in Table 1. They show that the number of "solitary" tracks (i.e the convection cells that do not interact with others by means of merging and splitting), and their contribution to total rainfall, is lower for higher resolution. Vice versa, the total contribution of the tracks that undergo merging and splitting is clearly higher for the

better resolved simulations. From this observation we draw the conclusion that there is more clustering happening at the higher model resolutions. The radar results looks somehow in between, with more clustering than in the 625 m model result, but less than in the finest resolution. We will clarify this in the revised manuscript.

---

## Author Response (AR1)

Dear Ms Töpfer, dear referees,

Thank you for considering our manuscript. In this file, we provide our replies to the referee comments, along with the revised manuscript. The original referee's comments are written in black, and our replies are highlighted in blue color.

In the highlighted manuscript file, we have marked all changes and additions to the text in blue color. Small corrections are not highlighted. Fig. 3 is new, and minor changes have been made to Figs. 2, 4, 8, and 9. We have also made small additions to the Acknowledgement section (highlighted).

In response to the suggestion by referee #1 (comment no. 1), we have changed the title of the manuscript to: "Impact of resolution on Large Eddy Simulation of mid-latitude summer time convection". Further, we have added the following references:

- Langhans et al. (2012)
- Neggers et al. (2019)
- van Stratum and Stevens (2015)
- Ban et al. (2014)
- Ban et al. (2015)
- Kendon et al. (2014)
- Louis (1979)
- Emori and Brown (2005)
- Pfahl et al. (2017)

We hope that you find the manuscript appropriate for publication in ACP in the present form.

Sincerely,
Christopher Moseley (in behalf of the authors)

**Referee #1**

This article discusses the impact of resolution on the organisation of convection in a LES of summer time convection over Germany, as well as the sensitivity of precipitation to 2m temperature in simulations with 625m grid spacing.

It concludes that there is a benefit in using a simulation with 156m grid spacing as compared to 625m in terms of the diurnal cycle of convection and some of the measures of convective organisation, and that the model underestimates the sensitivity of rainfall to 2m temperature.

Most of the analysis is a valuable analysis of ICON-LEMs representation of summertime convection. I have some questions about both the methodology and the conclusions, and some revisions will be required to make the manuscripts suitable for publication.

The writing is mostly clear, although some of the sentences are rather long and the

language could be more concise at points (I have suggested some changes here, but more could be made). There are also places where e.g. including hyphens would make the text more readable.

Thank you very much for carefully reading our manuscript, and for your detailed feedback. For easier readability, and for referencing, we have attached numbers to your major comments. Below are our replies to your individual comments.

General comments:

1. Title: "resolution and air temperature" -> I find this a bit confusing, as resolution is determined by the model configuration, but air temperature is not a model parameter (it impacts the simulated convection, rather than the simulation itself). Maybe mention "sensitivity to 2m temperature" specifically?

We changed the title to: "Impact of resolution on Large Eddy Simulation of mid-latitude summer time convection", thus leaving out the temperature sensitivity.

2. It would be good to add some further information about earlier studies that have looked at sensitivity of convection to resolution. One term that has come up in recent years is so-called bulk-convergence (i.e. the convergence of larger-scale mean properties) as opposed to structural convergence (e.g. Langhans et al 2012, https://journals.ametsoc.org/doi/full/10.1175/JAS-D-11-0252.1, Panosetti et al 2019, https://rmets.onlinelibrary.wiley.com/doi/full/10.1002/qj.3502).

Thank you for drawing our attention to these studies. Panosetti et al. (2019) write that they find neither bulk-convergence nor structural convergence over Germany at the 1 km grid spacing scale. Their highest resolution simulation has 550 m grid spacing which is close to our outer ICON-LEM nest with 625 m grid spacing. Although we analyze all simulations on the same grid, our study mainly addresses structural convergence, as it is mainly concerned with the shape, time evolution, and organization of individual convection cells, rather than mean quantities averaged over areas that are larger than individual clouds. Our conclusion is that there is no structural convergence at grid spacings that are coarser than the 100 m scale, which is consistent with Panosetti et al.. In the revised manuscript, we mention the concept of bulk- and structural convergence in the Introduction (p.2 l.34 ff), and refer to it in the discussion of our results (p.21 l.7).

3. p4, l26: The authors mention they have resampled their results on a larger grid. Although such a resampling is a good idea, it is important to be aware that the method used may influence the results. For example, it is likely that the cloud fraction increases due to the resampling, because some grid cells will only partially meet the threshold (this is certainly the case if non-zero liquid water would be used as the mask). It is not fully clear to me how this can be prevented, but it may be worth describing the possible effects. One alternative strategy for regridding would be to randomly sample one of the columns: this would keep the cloud fraction the same (statistically). This may also be relevant to the track statistics (section 3.3.)

There are several reasons why we applied a regridding. The main reason is that the original ICON-LEM output is given on an unstructured triangular grid. The codes for the calculation of the indices, and the rain cell tracking, need a regular lat-lon grid as input, and an extension of our codes to handle the unstructured raw data would be a difficult task. The second reason is that we prefer to compare the data of the three different model resolutions, and the radar data, on the same grid, to reach a fair comparison. We chose a 1x1 km lat-lon grid, since this is roughly the resolution of the radar data. Further, it is only slightly coarser than the resolution of the coarse ICON-LEM resolution with 600 m grid spacing of the triangle edges. However, as the **effective** resolution of the ICON-LEM data is larger than the grid spacing, we can assume that there is no loss in resolution at least for the 600 m simulation. A similar regridding has also been used for other studies which also analysed ICON-LEM output, like Heinze et al. (2017), and Pscheidt et al. (2019). We mention this in Sec. 2.3 (p.6 l.12 ff).

As you point out, there could be some problems that the cloud edges are not clearly defined, since the resampling may lead to lower values such that the threshold may not be reached at some grid boxes. However, since we track the surface precipitation field, with a relatively low threshold of 1 mm/h rain intensity, we assume that this effect is rather small at a resolution of 1 km. In Moseley et al. (2013) (https://doi.org/10.1002/2013JD020868) it was shown that at least for radar data over Germany, the rain cell tracking result is not largely different if the resolution is 1 km or 2 km. The conservative remapping method also takes care that the total amount of rainfall is not changed, which is not the case with a random sampling of columns.

4. Section 2.4: It would be good to add some more information on the interpretation of and differences between some of the indices of convection, such as SCAI and COP. It is not clear to me what the advantages of using one metric over the other would be from the current description.

A more detailed discussion of the organization indices and the differences between them is given in Pscheidt et al. (2019). This study uses the same simulation domain as our paper, and partly the same data. Therefore we can refer to it for more information on the indices, their general behavior over the study domain, and how they should be interpreted. The purpose of our present study is mainly to show which of these indices are affected mostly by model resolution, and investigate their sensitive to daily mean temperature. Another main purpose of our study is to show that the rain cell tracking method can add additional information on the temporal evolution of convective organization that the indices can not provide, since they only "see" the spatial distribution of convection at instantaneous time records. For these reasons we decided to calculate and present all four indices SCAI, COP, I.org, and I.shape in the paper.

In their conclusions Pscheidt et al. write that since COP and SCAI are mainly influenced by the areas and the number of objects, respectively, they recommend to use I.org, I.shape, object area, and object number for characterizing the state of the spatial organization of the convection field. They also claim that I.org is in some respects superior to COP and SCAI, since it is able to distinguish between three possible categories: Organized, regular, and random. We added the following lines to the Discussion section (p.22 l.4 ff):

"Pscheidt et al. (2019) recommend that COP and SCAI can be replaced by object sizes and object number, respectively, since they are mainly influenced by these two quantities. However, supplementary information on the degree of organization is provided by Ishape and Iorg, in particular since the latter is able to distinguish between three possible categories: Organized, regular, and random. Our study confirms this hypothesis, with the addition that tracking objects in time can give valuable information on the tendency of convection to form clusters. "

5. One of the metrics which is currently missing, and may be helpful in terms of the interpretation of the other indices, is a probability distribution function of object sizes in each simulation and the radar. This could potentially be plotted both for the original data and the resampled data.

We have plotted PDFs of precipitation cell sizes for the three different model resolutions, and for the radar data, respectively. As we did not save the original data but only the regridded data-sets, we can perform this analysis only on the regridded data. The following figure shows the cloud size distributions, including all three 3-domains days between 6 and 21 UTC:

[Figure]

Panel (a) shows the normalized size distribution, and (b) shows the (un-normalized) total number of detected cells. It can be seen that the RADOLAN data show a larger fraction of large objects, but fewer small objects that can be attributed to isolated cells, compared to the DOM01 (625 m) nest. However, the *total* number of large cells in the radar data is not much different from the simulations. For the higher resolved nests, the fraction of small objects is closer to radar. We have included the above figure into the manuscript (new Fig. 3) and added a paragraph in Sec. 3.1 (p.9 l.1 ff). This picture is consistent if the size PDF is plotted for each of the days individually (we did not include these plots into the paper):

[Figure]

6. One potential issue with some of the metrics, e.g. Iorg, may be that it can give

disproportionally high importance to smaller objects. One option here would be to consider a measure of organisation that considers objects of the same size (see e.g. Neggers et al 2019, https://journals.ametsoc.org/doi/full/10.1175/JAS-D-18-0194.1). It would be good to mention this in the text.

Thank you for recommending this interesting study. Neggers at al. write that spatial organization affects both ends of the precipitation cell size PDF, but in different ways: While the number of large clouds increases, there is an enhanced variability in the number of small clouds. This suggests that it is probably possible to extract more information on the spatial organization of convection, if new indices can be defined that take into account smaller and larger clouds separately. However, in our study, we are mainly concerned with deep precipitating convection where cloud sizes below 250 m are neglected, while Neggers at al. look also at smaller shallow cumulus clouds.

We have mentioned this possible shortcoming of existing indices in the Discussion section in the revised manuscript with reference to the study by Neggers at al. (p.22 l.7 ff). However, the improvement of existing indices for the spatial distribution of convection is not the main purpose of our study (see reply to your comment 4), and a subsampling analysis similar to Neggers et al. would be beyond the scope of the paper.

7. In section 3.4, there seems to be a significant difference between all simulations and the radar in terms of the organization indices. It would be good to investigate the cause of this in more detail, for example by looking at object size distributions, or the original fields from which the indices were derived.

Is your question referring to Fig. 3 (as there is no section 3.4 in the manuscript)? As stated in our reply to your comment 5, we plotted size distributions for the 3 resolutions and the radar data. Regarding the differences in the indices between model simulations and radar data in Fig. 3, the plotted size distributions seem to be consistent at least with the SCAI and COP index (see also our reply to your comment no 4): The smaller total number of objects in the radar data is reflected by the reduced SCAI (especially in comparison to ICON 600 m), and the smaller number of small objects in the normalized distribution is consistent with the larger value of COP. Although the size distribution does not provide any direct information on the shape of objects, the smaller value of I.shape in the radar data is consistent with the larger fraction of large objects, since large objects are more likely to deviate strongly from the circular shape. We have included this explanation in section 3.2 (p.9 l.10 ff).

8. One striking feature of figure 3 is that the development of SCAI looks different between different days. The other metrics seem to have a very similar development on different days, and for COP and I-org, the differences between radar and simulations are of the same order as the differences between the development of the indices on different days. This may point to the SCAI being more useful than some of the other indices.

As we mentioned in our reply to your comment 4 with reference to Pscheidt et al. (2019), SCAI is mainly influenced by the number of objects. The strong differences in SCAI between different days thus hint to different numbers of convection cells on the different days. On the other hand, the observation that COP and I.org are more robust among the days could reflect the fact that size distributions do not vary that much among days (see the size PDF

plots for the individual days as shown in our reply to your comment no 5). We added the following paragraph in Sec. 3.2 (p.10 l.28 ff):

"An interesting observation is that SCAI differs more strongly between the days, while for the other indices, the differences among the simulation nests and the radar data are of the same order as the differences between different days. The reason could be that SCAI follows closely the total number of rain cells which varies strongly between days, while the other indices are rather linked to the size distribution which is similar on all days."

In our reply to your question no 4 we have argued that SCAI could therefore simply be replaced by the total number of objects. Thus our conclusion argues rather against the usefulness of SCAI (and COP) than in favor of it. We hope that this point is now clearer in the revised manuscript.

9. One aspect of SCAI that I am wondering about is the fact that it seems to be consistently low at night. This may partially be due to organised propagating systems, but I am also wondering how the SCAI behaves when convection is (almost) absent? Is there a strong correlation between SCAI and cloud cover?

As SCAI is strongly affected by the number of objects, and the number of objects is larger when there is strong convection, it seems plausible that SCAI is larger at noon time when convection sets it. At night, precipitation is mainly large-scale, with larger, but fewer objects and weaker intensities. SCAI already begins to decline in the afternoon hours, which reflects the observation that convection begins to organize and forms larger objects due to merging, thus reducing the number of objects.

We did not store the cloud cover field fields, but we checked the correlation between SCAI and cloud liquid water (LWP) field. We plotted LWP into Fig. 2, and a comparison with Fig. 3 shows that SCAI follows more closely the mean precipitation intensity than the mean LWP. We have mentioned this in section 3.2. (p.10 l. 4-5).

However, we point out that our study focuses on daytime convection only, since it is known that the nocturnal boundary layer is not sufficiently resolved at LES resolutions of 100 m and coarser, which may introduce unknown biases in cloud cover at night. See e.g. van Stratum and Stevens (2015) https://doi.org/10.1002/2014MS000370. We have included a citation of this paper in Sec. 2.1 (p.4 l.11 ff).

10. What explains differences in night-time behaviour between cool and warm days?

As stated in our previous reply, we cannot make any statements on night time precipitation from our LES simulation, since the nocturnal boundary layer is unresolved. Therefore we condition or analysis on daytime temperatures between 8 and 20 UTC, only. To make this clearer, we have limited our plots to the time range between 6 and 21 UTC in the revised manuscript.

11. The results should likely be interpreted in the context of a given configuration. It would be worth stressing that changes to e.g. the microphysics scheme, as well as further changes to the turbulence scheme mentioned already, will impact on the results. - In

figure 8, again there seem to be differences between radar and ICON in terms of Ishape and COP, which are bigger than the differences between warm and cool days.
Do you understand what causes these differences?

As we neither have sensitivity studies with different turbulence schemes, nor with changed microphysical parametrizations, we cannot make any statements on the impact of different schemes on our results. We agree that there might be significant impacts on organized convection. We have added the following sentences to the Discussion section (p.21 l.23 ff):

"We also note that the microphysics scheme might have significant impacts on organized convection. An analysis of the impact of different physical parametrizations on the simulated convection is not covered by this study, and we encourage future studies in this direction."

The lower COP and higher I.shape in the model data as compared to radar hint at an underrepresentation of convective organization and more compact objects in the 625 m LES, respectively. However, radar and model agree that organization is stronger on warmer days. We have rephrased the last paragraph in section 4.2 accordingly (p.17 l.8-10). The reason for the suppressed organization could be related to the too explosive convective initiation at coarser resolutions. As soon as a convection cell is initiated, it is already fully developed and does not have enough time to interact with neighbouring cells within its life time. However, this is a hypothesis that should be tested in a future study. Such a study should investigate the processes that happen within merging cells more deeply. We have mentioned this in the Discussion section (p.21 l.18 ff).

12. p19, l11: "larger clusters". I am not sure if this can be said on the basis of the statistics provided. Can you clarify?

This statement mainly refers to the interpretation of the tracking analysis, notably the numbers shown in Table 1. They show that the number of "solitary" tracks (i.e isolated convection cells that do not interact with others by means of merging and splitting), and their contribution to total rainfall, is lower for higher resolution. Vice versa, the total contribution of the tracks that undergo merging and splitting is clearly higher for the better resolved simulations. From this observation we draw the conclusion that there is more clustering happening at the higher model resolutions. The radar results are somehow located in between, with more clustering than in the 625 m model result, but less than in the finest resolution. We have re-written this passage in the Discussion section (p.21 l.10 ff):

"In contrast, at 156 m, we find a smoother onset of convective updrafts with lower peak intensities, and a stronger degree of organization, that in general show a better match with the radar data. In addition, the tracking analysis revealed that the stronger organization of the higher resolved simulations is accompanied by an increased tendency of convection to form larger clusters: The 156 m simulation shows a lower number of isolated rain cells, and their contribution to total rainfall is lower. Vice versa, the total contribution of the tracks that undergo merging and splitting is clearly higher for the higher resolved simulations."

———————————————

Minor/editorial issues (these are mostly easy to address, but could improve the presentation quality):

- p1, l4-5: "showing a considerable..most of the days" -> this is a long clause, maybe it can be broken up?

Rephrased in two sentences.

- p1, l8: "showed that"

Corrected.

- p1, l14: diurnal cycles -> "the diurnal cycle" is clearer, I think.

Changed.

- p1, l16: "CRMs" (plural)? Or refer to the technique instead.

Changed to plural.

- p1, l16: "necessary" -> I would simply say "suitable", possibly a well-designed parametrisation could also have the correct diurnal cycle.

"Necessary" removed.

- p1, l21: it would be good to explain the differences between LES and kilometre-scale modelling in terms of the turbulence scheme.

We have added in on p.1, l. 20 ff: "Regional limited area models allow for even higher resolutions with grid spacings in the sub-kilometer range with Large Eddy Simulations (LES) where the large eddies of the turbulence spectrum are modeled explicitly as opposed to a fully parametrized turbulence spectrum in the convection permitting simulations."

- p2, l1: this sentence is on the long side.

Sentence shortened.

- p2, l10: the cumulus scheme would be worth mentioning here as well.

Here we specifically refer to CRMs *without* cumulus parametrization. To make this clear, we now explicitly write this on p.1, l. 16/17.

- p2, l16: the presence of super-CC scaling may depend on the method of analysis (Ban et al. 2015, https://agupubs.onlinelibrary.wiley.com/doi/full/10.1002/2014GL062588).

We have cited this paper in the Introduction (p.2 l.19): "Although these studies have applied different methods to determine the temperature scaling rate that can lead to different results (Ban et al., 2015), the super-CC scaling seems to be a robust feature that has been found by several studies for present day climate."

- p2, l16-17: "even" occurs twice here.

Corrected.

- p2, l20: there are earlier references on the dynamic and thermodynamic components of this sensitivity.

We have added the references Pfahl et al. (2017), and Emori and Brown (2005). We have added in the manuscript (p. 2,l. 22 ff):
"Analyses of climate change projections have indicated that while the thermodynamic contribution to the intensification of extreme precipitation is expected to be relatively homogeneous globally, there may be strong regional differences in the dynamic contribution due to changes in circulation patterns (Emori and Brown, 2005; Pfahl et al., 2017; Norris et al., 2019)."

- p2, l22: "air temperature" is a somewhat vague term. I would probably mention 2m temperature specifically, unless a different level/set of levels is used.

Change to "2m air temperature"

- p2, l24: "above-mentioned"

Corrected.

- p2, l27: "object-oriented"?

Corrected (2 occurrences).

- p3, l5: "by" -> "of"?, "suited to provide" -> "provides"

Corrected.

- p3, l9: remove "applied"

Removed.

- p3, l10: 165 -> 156?

Typo, corrected.

- p3, l15: remove "implemented".

Removed.

- p3, l22: place a comma after "work".

Done.

- p3, l26: it may be good to mention something about the surface layer parametrization and the (absence of?) a subgrid-scale cloud scheme. For the turbulence scheme, it would be good to mention how grid anisotropy is dealt with.

We have added on p.4, l. 2 ff: "We only emphasize that turbulence is parametrized using a Smagorinsky model (Dipankar et al., 2015) (thus, subgrid turbulence is treated as isotropic), the land surface is described using the TERRA-ML model (Schrodin and Heise, 2002), the surface layer is treated with a drag-law formulation following Louis (1979), a simple all-or-nothing cloud scheme is used} and gravity waves (orographic and non-orographic) are not parametrized."

- p3, l29: "down scaled" -> I am not sure if this is the right verb.

We think that "dynamical downscaling" is the appropriate term for the nesting approach that was applied in the ICON-LEM simulations. We rephrased (p.4 l.8):
"Dynamical downscaling in a one-way nesting approach is applied on 3 of the model days, in a first step to 312 m, and in a second step to 156 m grid spacing (Heinze et al, 2017)."

- Section 2.2: it would be good to spend some text on model initialisation and spin-up of convective structures for the inner nests.

We have added the paragraph (p.5 l.7 ff):
"In all simulations, the state of the atmosphere and the soil has been initialized at 0 UTC with COSMO-DE data. The first 6 simulation hours are used as spin-up for the atmosphere, and are removed from the analysis. For the high resolved 3-domain simulations, all 3 nests are initiated at the same time."

- p4 l4: article "the" missing before billions

Corrected.

- p4, l5: do you mean "the first days"?

Yes. Corrected.

- p4, l8: comma after "costs"

Corrected.

- p4, l11: the differences are also partially due to the inherent unpredictability of convection

We have added this on p.4 l.26.

- p4, l14: "wide spread" -> "widespread"

Corrected.

- p4, l14: can you give some more information on the presence of cold pools during these days.

Cold pools are clearly visible on all convective model days. Currently, there are follow-up studies in preparation that analyse cold pools in these simulations. Therefore, we decided not to mention cold pools in this paper.

- p4, l23: "large scale" -> "large-scale"

Corrected (this sentence has been moved above the bullet points and put in parentheses).

- p5: l4 "time-interpolated"?

Corrected.

- p6, l25: note that I-shape is sensitive to discretisation: for example, as far as I can tell, a circle that is approximated by a large number of squares would have a shape ratio of pi/4, rather than 1.

This problem may rather show up if there is a large fraction of small objects which are not properly resolved. As we cannot provide a detailed analysis of the sensitivity of the indices to horizontal resolution, we decided not to discuss this in the paper. However, we think that it would not significantly affect the general qualitative behavior of I.shape in our study.

- p6, l27: "contour"

Corrected.

- p7, l4: "defined...results" -> this a very long clause, it would be better to split it.

Sentence broken up.

- p7, l31: "different resolutions" (plural)

Corrected.

- p7, l32: "the" (capitalisation)

Corrected.

- p8, l1: are you referring to June 6, instead of June 3, here?

We referred to June 3, but the sentence was probably not very clear. We rephrased (p.8 l.24): "Especially on June 3, both the magnitude and the timing of the precipitation peak is closer to the radar data for the higher resolved domains than for the 625 m domain."

- p8, l1-5: the later onset in the simulations with higher resolution appears to be consistent on June 3 and June 6. However, some of the other differences may be due to

individual large storms. It would be good to mention this at least (running ensemble forecasts of the lowest resolution run would help to establish this internal variability, though I am aware this may be a major effort).

Yes, running ensemble forecasts would be a major effort that we cannot carry out within this paper revision. But we mentioned on p.8 l. 28 ff: "We note that although the later onset in the simulations with higher resolution appears to be consistent on June 3 and June 6, we cannot rule out that some of the other differences may be due to internal variability, like individual large storms."

- In section 3, the terms organisation and clustering are used somewhat interchangeably (it may be good to make explicit which of the measures identify clustering in particular, I thought this was mainly I-org).

The terms are not completely interchangeable. Rather, we understand clustering as a special *type* of organization. For example, a regular distribution of objects would be organized, but not clustered. SCAI and COP mainly measure the degree of clustering, but I.org is also able to detect a regular, but non-clustered, organization. To make this clear, we write in Sec. 2.4 (p.7 l.5):

"Unlike SCAI and COP, which mainly quantify the degree of clustering, the NN-based organization index I.org (Tompkins and Semie,2017) is able to distinguish between three types of spatial distribution: clustered, regular, and random."

In our simulations and radar data, we find clustered organization, therefore we frequently use the term clustering.

- p8, l16: "somehow different" -> remove "somehow", explain the differences for June 6.

The explanation follows later in the text. We shifted the mentioning of the differences on June 6 to the appropriate position (p.10 l.21):

- Caption fig. 3: 165 -> 156

Corrected.

- p9, l10: "We now how...tracks" -> I would rephrase this sentence, to tell more about the kind of additional information provided, rather than the fact that additional information is provided.

Sentence removed. The additional information in discussed mainly in the Discussion section.

- p9, l10: "are provided" -> "is provided"

Obsolete (previous comment).

- p10, l2: there is an issue with the parentheses here.

Corrected.

- p10, l5 "even" -> "event"

Corrected.

- p10, l15: remove italics here (m) for consistency (same applies to e.g. p11, l16 and p13, l1/12/18)

Done for all occurrences.

- p10, l17: could the relative percentages be affected by the regridding method?

As we have explained in reply to your major comment no 3, the regridding could affect the object identification. However, as the detection of merging and splitting events is also not completely un-ambiguous, we expect that the impact of the regridding on the tracking result is minor.

- p10, l20: "composites"

Corrected.

- p10, l24: "sizes" (capitalisation)#

Corrected.

- p11, l1: "at" -> "in"

Corrected.

- p11, l4 "(g-i)"

Fig. number was wrong (Now Fig. 5). Corrected.

- p11, l9: it would be good to add a subscript to the areas, and put "A" in italics.

Done.

- p11, l20: "including" -> "and for"

Changed.

- p11, l30: it would be good to refer back to the concept of bulk-convergence here.

We have mentioned bulk convergence and structural convergence on p.14 l.19.

- p12, l8: in terms of differences between precipitation between model and forecast, some of these may be due to the uncertainty in boundary conditions.

We have added on p.15 (bottom lines): "We note that in addition to these systematic differences, some of the differences between model and radar data could also be traced back to the uncertainty in boundary conditions from the COSMO forecast data."

- p13, l4 "added value" (no article)

Corrected.

- p13, l10: "6-day period"

Corrected.

- p13, l14: "introduction" (capitalisation)

Corrected.

- p14, l9: "than for the mean"

Corrected.

- p14, l12: "introduction" (capitalisation)

Corrected.

- p14, l14: is an underestimation of sensitivity to temperature sensitivity consistent with previous findings? It is not clear to me if this result is significant, given that only a few rain cells may have a big impact on the 99th percentile rainfall. The sensitivity test here is not a very strong one, as the bulk of the underlying data stays the same. One option would be to look at how much this differs between (subsets of) individual days in each category (looking at the diurnal average 99th percent rainfall).

To directly compare the simulated temperature sensitivity of heavy precipitation with observations, as we have done in our study, one requires a cloud resolving model simulation of a longer time period in a realistic setup over an area that is covered by observations. Given the high computational demand of these simulations, such studies still seem to be rare. The only study we have found that met these requirements is Ban et al. (2014) (https://agupubs.onlinelibrary.wiley.com/doi/full/10.1002/2014JD021478). They analyze the temperature scaling of a decade long simulation over Switzerland, and find a good agreement with observations at 2.2 km grid spacing (their Fig. 13). However, the strong orography in their study region is absent in the largest part of our simulation domain, such that a direct comparison to our study may be difficult. We have cited this study in the Introduction (p.2 l.19-21), and added in the Discussion section (p.22 l.32 ff):

"Although, in contrast to our results, Ban et al. (2014) do not report of an underestimated temperature sensitivity of heavy rainfall in Switzerland with a 2.2 km model, the strong orography in their study region is absent in the largest part of our simulation domain, such that a direct comparison to our study may be difficult."

To address the second part of your question, we performed the sensitivity test that you have suggested: We randomly chose 3 out of the 6 warm days, and 3 out of the 6 cool days, and reproduced the plot in Fig. 8 with these days. Then we took the 3 other warm, and respectively, cool days and again reproduced the figure (panels a-c). We repeated the whole procedure a second time, finally arriving at 4 reproductions of Fig. 8, each presenting a different subset of warm and cool days (we did not include these plots into the manuscript):

- Cool days: 06-12, 06-15, 06-17: Warm days: 06-04, 06-05, 06-25

- Cool days: 06-13, 06-14, 06-19; Warm days: 06-06, 06-07, 06-29

- Cool days: 06-12, 06-15, 06-19; Warm days: 06-05, 06-07, 06-29

- Cool days: 06-13, 06-14, 06-17; Warm days: 06-04, 06-06, 06-25

[Figure]

Although the mean precipitation intensities (panels a) and the mean water vapor path (panels c) differ quite strongly between the 4 cases, the 99th percentile (panels b) consistently shows the following two features:

1. Peak intensities are stronger (weaker) for warm (cool) days in both radar and model
2. The difference between warm and cool days is weaker in the model than in radar

This confirms our hypothesis that ICON-LEM at 600 m grid spacing underestimates the scaling of extreme precipitation with temperature. We added in section 4.1 (p.16 l.22 ff):

""As a sensitivity test, we randomly chose 3 out of the 6 warm days, and 3 out of the 6 cool days, and reproduced the plot in Fig. 8b with these days (not shown). Repeating this procedure 4 times confirmed that peak intensities of the 99the percentiles are stronger (weaker) for warm (cool) days in both radar and model data, and second, that the difference between warm and cool days is weaker in the model than in the radar data.

- p15, l10: "similar to the larger period" -> "similar to that in the larger period"

Corrected.

- p16, l1: "a stronger degree"

Corrected.

- p17, l2: "there" (capitalisation)

Corrected.

- p17, l2: the trend is consistent, but the actual number of solitary tracks is very different. The differences between "cool days" and "all days" also seem more pronounced in the radar data.

The difference in the trend of the fraction and the total number of solitary tracks is discussed later in section 4.3. Your observation that the trend is more pronounced in the radar data is correct, we mention this on p.18 l.5. It is also mentioned at the end of the section.

- p17, l3/l5/l9: "there","instead","while" (capitalisation)

Corrected.

- p17, l4-19: may this impact on the interpretation of the 3-domain days as well (in the light of the remark about regridding in the general introduction)?

This is a good point. We think that the interpretation in the case of the 3-domain tracking analysis is simpler, since the differences in the total number of tracks between the domain resolutions are not as large as in the temperature sensitivity analysis. We have added the following sentences to the manuscript:

Section 3.3 (p.11 l.3-5):"In total, the algorithm detects 141682 tracks for DOM01, 160042 tracks for DOM02, and 124820 for DOM03, showing no clear trend with resolution. For the radar data, a smaller number of 67657 tracks is detected."

Section 4.3 (p.19 l.9-11): "We note that the differences in the track statistics between warm and cool days have a different quality than the differences between the model resolutions as found in Sec. 3.3, since the differences in the total number of tracks among the resolutions are smaller."

Discussion (p.22 l.23-25): "The large differences in the total number of tracks between warm, cool, and all days in the analysis of the temperature sensitivity makes the interpretation of the tracking result more difficult, compared to the resolution analysis."

- p17, l20: "Similar as" -> replace by a construction with "similar to"

"Similar as Fig. 6" removed.

- p17, l24: "temperaure" -> "temperature"

Corrected.

- p18, l2: "the longer durations tracks above 1 hour life time" -> "the tracks with a life time longer than 1 hour"

Changed.

- Figure 5 and 9 have intensity with units km. This should likely be mm/hr.

Yes. Figures corrected.

- p18, l4: "on warm days" is repeated here.

Corrected.

- p19, l4: see my comments on the title. These are different kinds of sensitivities (to the model configurations, to the atmospheric state).

We have rephrased the first sentence of the Discussion: "We have evaluated the impact of horizontal resolution on explicitly simulated convective precipitation, and analysed the sensitivity of convective organization to daily mean 2m air temperature on the 36 day continuous simulation with 625 m grid spacing."

- p19, l15: it could be good to cite some work on turbulence schemes for the boundary layer
grey-zone here.

As we do not explicitly discuss the gray zone in this paper, we prefer to merely mention that the turbulence scheme and the microphysics scheme also have an impact and is not discussed here.

- p19, l18: "should be left" -> "will be left"

This passage is re-formulated.

- p19, l19: "similar as" -> "similar to"

Corrected.

- p19, l20: "fewer and larger objects"

Corrected.

- p19, l23: "as to compared to" -> "as compared to"

Corrected.

- p19, l31: "at least simulated qualitatively" -> it would be good to rephrase this (the word choice/order here is odd: you could say the sensitivity to temperature has the same sign)

We have rephrased: "Consistent with theory, our analysis of the continuous 36-day period with 625 m grid spacing shows that convection gets more intense with higher near-surface temperatures."

- p20, l3: "we","in" (capitalisation)

Corrected.

- p20, l7: "objects" (plural)

Corrected.

- p20, l8: "may be left" -> "is left"

Corrected.

- p20, l11: "higher-resolved" -> or change formulation?

Rephrased (p.22, l.27-29): "Our study also cannot answer the open question if higher resolution will lead to an improved simulation of the sensitivity of heavy rainfall and convective organization to temperature, as too few high resolved model days are available."

- p20, l28: "in May and June 2016" -> "for May and June 2016"

Corrected.

- p20, l30: but to a smaller degree. -> the sentence construction needs to be changed here (see also my previous comment on significance).

Rephrased (p.23 l.13-15): "Based on a 36 day long continuous simulation for May and June 2016, we have shown that ICON in a limited area setup over Germany and a grid spacing of 625 m is able to simulate an intensification of isolated convective rain cells with temperature. However, the magnitude of the simulated intensification is smaller than shown by the RADOLAN radar composite."

- p21, l1: Remove "However"

Removed.

- p21, l5: place a comma after "Overall".

Done.

**Referee #2**

This paper examines the convection activity simulated by the ICON-LEM at 625 m horizontal resolution for 36 days in summer over Germany and compares it to that observed by the ground radar system. It also examines the impact of horizontal resolution on the simulated convection over 3 days by nesting into 300 and 100 m resolutions. I agree with the other reviewer that the authors showed great expertise in terms of deployment of the model and the analyzing techniques. But the logic of the paper, in other words, why they set up and present the study in such a way to answer the questions they want to answer, is not clear enough to me. I suggest a major revision. Here are my major concerns:

Thank you for your comments and suggestions for improving the manuscript. Please find our replies to your comments in blue color.

1. In the analysis of the 36-day simulations, the authors separated the cases into warm and cool days to compare the impact of surface air temperature on convection activity as seen in simulations and in observation.One of the motivations for doing this seems to be that the authors are concerned with the ability of the cloud-resolving models to correctly simulate the response of atmospheric convection in a warming climate, which they hinted at in the introduction. But it is not obvious to me that the contrast in the large-scale environment between the warm and cool days chosen in this study is comparable to the contrast in typical large-scale environment of middle-latitude convection activity between the current and future warmer climate. If the authors think they are comparable, they should make the claim more explicitly. If the authors just want to compare the sensitivity of the simulated convection to different environmental conditions as characterized by the surface air temperature the

analysis is completely valid in my opinion given the importance of surface condition for summertime convection over mid-latitude land.

You are correct that our study is not able to predict changes in convective precipitation under climate change. Thank you for drawing our attention to this possible misunderstanding. The practice to investigate the sensitivity of heavy precipitation to temperature by conditioning high percentiles of precipitation intensity on daily mean temperatures has been originally proposed by Lenderink and van Meijgaard (2008), and has been adopted by several subsequent (mainly observational) studies. Many of these studies refer to climate change as a motivation, reasoning that warmer temperatures in the future are likely to produce higher precipitation extremes. In our manuscript, we also follow this approach based on the 35 simulation days that we have available. Of course, changes in large-scale circulation and variability also have to be taken into account when making statements about climate change, but they are not considered here. We have rephrased the Introduction section in the revised manuscript to make this clear (p.2 l.12-14). We have also mentioned climate change studies that have analysed extreme precipitation in climate change projections (p. 2, l. 22-25):

"Analyses of climate change projections have indicated that while the thermodynamic contribution to the intensification of extreme precipitation is expected to be relatively homogeneous globally, there may be strong regional differences in the dynamic contribution due to changes in circulation patterns (Emori and Brown, 2005; Pfahl et al., 2017; Norris et al., 2019)."

In addition, first climate change studies like Kendon et al. (2014) have been published, that have analyzed the intensification of extreme precipitation in a future warming scenario with a cloud resolving model. We have cited this study in the Introduction (p.2 l.29). Please note also our new citation of Ban et al. (2015) that we included in response to Reviewer #1 (p.2 l.20), which argues that an extrapolation of present day temperature scaling into the future is problematic.

2. I wonder what criterion the authors used for selecting cases to perform higher-resolution simulations and why the authors did not choose those cases so that they could also investigate the contrast between warm and cool days at higher resolutions(even just with 2 or 4 cases).

Given the high computational costs of such simulations, especially the high resolution simulations with 3 nests, we had to constrain our analysis to the simulated days that were available. We chose the given 35 days period as there was convection in a large part of the domain in almost all of these days, but due to the available computing time and storage space we could only run it on the 625 m nest. 3-nest simulations were performed for pre-selected days within the German joint project "HD(CP)$^2$ High definition clouds and precipitation for advancing climate prediction" (mentioned in the Acknowledgement) of which our study was part of, and out of 4 available high resolved model days within this period, we chose 3 for our analysis. As can be seen in Table A1, only one of these days (June 6) falls inside the "warm" category. None of the available 3 nest days are within the "cool" category. This is insufficient for an analysis of the temperature sensitivity for the higher resolutions, unfortunately. Therefore we decided to present the results of the resolution impact (based on

the three high resolved days), and of the temperature sensitivity (based on the 36-day simulation with 625 m) separately, and stated in the Discussion (p.22 l.27 ff):

"Our study also leaves the question open if higher resolution will lead to an improved simulation of the sensitivity of heavy rainfall and convective organization to temperature, as only three model days are available on the higher resolved nests. Given that the magnitude of the intensification of heavy rainfall with temperature has both a thermodynamic (based on the CC argument) and a dynamic aspect, and that thermodynamic processes can be expected to be rather independent of resolution, we can assume that it is mainly an insufficient representation of the dynamics within the convection cells that causes an underestimated intensification at 625 m grid spacing."

As our results show that convective life cycles and convective organization are better represented at the 100 m scale, we may speculate that also the sensitivity of heavy precipitation to temperature will be better simulated.

**Impact of resolution on Large Eddy Simulation of mid-latitude summer time convection**

Christopher Moseley[1,2], Ieda Pscheidt[3], Guido Cioni[1], and Rieke Heinze[1]

[1]Max Planck Institute for Meteorology, Hamburg, Germany
[2]Department of Atmospheric Sciences, National Taiwan University, Taiwan
[3]University of Bonn, Germany

**Correspondence:** Christopher Moseley (christopher.moseley@mpimet.mpg.de)

**Abstract.** We analyze life cycles of summer time moist convection of a Large Eddy Simulation (LES) in a limited area setup over Germany. The goal is to assess the ability of the model to represent convective organization in space and time in comparison to radar data, and its sensitivity to daily mean surface air temperature. A continuous period of 36 days in May and June 2016 is simulated with a grid spacing of 625 m. This period was dominated by convective rainfall over large parts of the domain on most of the days. Using convective organization indices, and a tracking algorithm for convective precipitation events, we find that an LES with 625 m grid spacing tends to underestimate the degree of convective organization, and shows a weaker sensitivity of heavy convective rainfall to temperature as suggested by the radar data. An analysis of three days within this period that are simulated with finer grid spacing of 312 m and 156 m showed that a grid spacing at the 100 m scale has the potential to improve the simulated diurnal cycles of convection, the mean time evolution of single convective events, and the degree of convective organization.

*Copyright statement.* The author's copyright for this publication is transferred to the Max Planck Institute for Meteorology, Hamburg, Germany.

[revised manuscript text omitted]

---

## Author Response (AR2)

**Second revision**

**Referee #1**

I am mostly happy with the revised manuscript: the authors have addressed the points I raised in a satisfactory way. My remaining comments are mostly small and editorial.

We thank the reviewer for final corrections and comments. We have implemented all mentioned suggestions, see our replies below. We have included two additional citations, in reply to the second comment:
- Haerter et al. (2019): Circling in on convective organization. *Geophysical Research Letters*, 46, 7024–7034
- Hirt et al. (2020): Cold pool driven convective initiation: using causal graph analysis to determine what km-scale models are missing. Under review at Q. J. Roy. Met. Soc.

- Please ensure colons are followed by lower case letters throughout the manuscript (maybe it is easiest to search for colon signs).

We have searched for colon signs as suggested, and corrected 8 occurrences.

- I think it would be good to mention a further analysis of cold pools is planned, as readers may find it remarkable these are not discussed in the text.

We have mentioned cold pools at the end of the Discussion section (p.23, l. 12 ff.), and have added the two references Haerter et al. (2019), and Hirt et al. (2020):

"Finally, we mention another important aspect that we have not addressed in the present study, but which however has been shown to have an important impact on convective organization, namely cold pools: when cold pool gust fronts collide, they sometimes trigger another convective precipitation cell, leading to a complex feedback between the convective rain cells and the cold pools that they generate (Haerter et al., 2019). Cold pools are clearly visible in our model data, and an analysis of their role in triggering new convection cells will be published in a separate paper (Hirt et al. 2020)."

- p1, l4: "This period was dominated by convective rainfall over large parts of the domain on most of the days." I think this wording is a bit confusing, as convective rainfall is restricted to a small part of the total domain at any point in time.

We have rephrased, to make clear that it is not the rainfall which covers a large part of the domain, but the convection (Abstract, l. 4):

 "This period was dominated by convection over large parts of the domain on most of the days."

- p1, l16: In general, some of these models may still have a parametrisation of smaller cumulus clouds, so I would say "without parametrization of deep cumulus convection", and state at another point that no cumulus parametrisation was used in the current study.

We have adopted this suggestion on p.1, l. 16, and have added in Section 2.1, p.4, l.4: "[...] and cumulus convection, as well as gravity waves (orographic and non-orographic) are not parametrized."

- p2, l1: "convection-permitting".

Corrected.

- p2, l3: It would be good to mention that at 625 m grid spacing, some of the larger boundary-layer eddies are likely unresolved (even if a Smagorinsky scheme is used).

We agree with the statement, but we find it more appropriate to mention it in the discussion section. We have amended (p.21, l.23):

"An improved subgrid scheme might lead to more realistic results and a decreased sensitivity to resolution, while the Smagorinsky subgrid scheme used in our model seems to be not the optimal choice at 625 m grid spacing, as some of the larger boundary-layer eddies are likely unresolved."

- p2, l12: "Assuming a warming trend in the future, and neglecting other influences like changes in large-scale circulation and variability that are attended by climate change, the sensitivity of precipitation extremes to warmer temperatures has been heavily discussed in the recent years." →The wording here is a bit confusing. You could start with "The sensitivity...years" and then continue with something like "Besides changes in global atmospheric temperature..."

We think that this passage may be more readable and less confusing when the other influences are not mentioned here, as we focus on the temperature increase in this context. That changes in large scale circulation and variability also play a role for climate change should be clear. We rewrote (p.2, l.13):

"The sensitivity of precipitation extremes to warmer temperatures has been heavily discussed in the recent years."

- p2, l20: To my best knowledge, Ban et al. find a scaling that is close to CC-scaling, rather than super-CC scaling (see the abstract of their study). They argue that the super-CC scaling can result from the imposition of e.g. thresholds for wet days in the data analysis. It would be good to double-check this.

After checking Ban et al. (2015) again, we agree with the reviewer's statement. We have rephrased the paragraph:

"In the meanwhile, several studies have found a super-CC scaling for present day climate. This indicates that beyond purely thermodynamic processes, also the dynamic component

within convective clouds contributes to the intensification and has to be evaluated separately. However, it has to be mentioned that some studies have found a scaling that is close to the CC rate, like Ban et al. (2015) who argue that the super-CC scaling might be an artifact that results from the statistical methods applied to determine the scaling rate, e.g. the imposition of thresholds for wet days in the data analysis."

- p2, l28: "decade-long"

Corrected.

- p2, l28: the wording may be a bit confusing here, do you mean the simulations use 2.2km grid spacing?

Yes. We have rephrased (p.2, l.29):

"Ban et al. (2014) have analyzed the temperature scaling of a decade-long simulation with 2.2 km grid spacing over Switzerland, and found a good agreement with observations."

- p2, l29: "have found" (plural)

Corrected.

- p2, l30: "However, a correct representation of the temperature scaling of heavy rainfall becomes increasingly difficult with decreasing model resolution, as Rasp et al. (2018) have shown that in principle subgrid cloud organization has to be included into stochastic cloud parametrizations." →" However, a correct representation of the temperature scaling of heavy rainfall becomes increasingly difficult with decreasing model resolution. Rasp et al. (2018) have shown that in principle subgrid cloud organization has to be included into stochastic cloud parametrizations. " (replace "as", which I tend to read as "because").

Corrected.

- p2, l35: "large-scale"

Corrected.

- p5, l8: "high resolved" → "high-resolution"?

Corrected.

- p8, l25: "higher resolver" → "with higher resolutions"

Corrected.

- p10, l21: "show" → "shows"

Corrected.

- p14, l19: I don't really think there is structural convergence in the life cycle statistics shown in Fig. 5. It is hard to judge if some of the statistics have converged without an additional simulation at even higher resolution.

We have weakened the statement (p.14, l.19):

"Although three model resolutions are insufficient to clearly identify bulk convergence and structural convergence, these results show an improved simulation of convection at the 100 m scale with ICON-LEM."

- p19, l10: "have a different quality" → do you mean they are more robust?

We mean that the tracking statistics between model resolution are more robust. We have rephrased (p.19, l.8):

"We note that due to the large differences in the total number of tracks between warm and cool days, the tracking statistics are more difficult to interpret here, compared to the more robust differences in track statistics between resolutions presented in Sec. 3.3."

- p21, l3: "36-day"

Corrected.

- p21, l11: "shows"? (singular)

Corrected.

- p21, l21: "more deeply" → "in more detail"?

Corrected.

- p22, l28: "high resolved model days": could you rephrase this (see suggestions above)?

Rephrased (p.22, l.30): "too few model days on all 3 nests are available"

- p22, l33: "report of" → "report"

Corrected.

- p23, l24: "and a grid spacing" → "with a grid spacing"

Corrected.

**Impact of resolution on Large Eddy Simulation of mid-latitude summer time convection**

Christopher Moseley[1,2], Ieda Pscheidt[3], Guido Cioni[1], and Rieke Heinze[1]

[1]Max Planck Institute for Meteorology, Hamburg, Germany
[2]Department of Atmospheric Sciences, National Taiwan University, Taiwan
[3]University of Bonn, Germany

**Correspondence:** Christopher Moseley (christopher.moseley@mpimet.mpg.de)

**Abstract.** We analyze life cycles of summer time moist convection of a Large Eddy Simulation (LES) in a limited area setup over Germany. The goal is to assess the ability of the model to represent convective organization in space and time in comparison to radar data, and its sensitivity to daily mean surface air temperature. A continuous period of 36 days in May and June 2016 is simulated with a grid spacing of 625 m. This period was dominated by convection over large parts of the domain on most of the days. Using convective organization indices, and a tracking algorithm for convective precipitation events, we find that an LES with 625 m grid spacing tends to underestimate the degree of convective organization, and shows a weaker sensitivity of heavy convective rainfall to temperature as suggested by the radar data. An analysis of three days within this period that are simulated with finer grid spacing of 312 m and 156 m showed that a grid spacing at the 100 m scale has the potential to improve the simulated diurnal cycles of convection, the mean time evolution of single convective events, and the degree of convective organization.

*Copyright statement.* The author's copyright for this publication is transferred to the Max Planck Institute for Meteorology, Hamburg, Germany.

[revised manuscript text omitted]